# The molecular basis for an allosteric inhibition of K⁺-flux gating in K₂ₚ channels

Susanne Rinné[1†], Aytug K Kiper[1†], Kirsty S Vowinkel[1], David Ramírez[2,3], Marcus Schewe[4], Mauricio Bedoya[2], Diana Aser[1], Isabella Gensler[1], Michael F Netter[1], Phillip J Stansfeld[5], Thomas Baukrowitz[4], Wendy Gonzalez[2,6], Niels Decher[1]*

[1]Institute for Physiology and Pathophysiology, Vegetative Physiology, University of Marburg, Marburg, Germany; [2]Centro de Bioinformática y Simulación Molecular, Universidad de Talca, Talca, Chile; [3]Instituto de Ciencias Biomédicas, Universidad Autónoma de Chile, Talca, Chile; [4]Institute of Physiology, University of Kiel, Kiel, Germany; [5]Structural Bioinformatics and Computational Biochemistry Unit, Department of Biochemistry, University of Oxford, Oxford, United Kingdom; [6]Millennium Nucleus of Ion Channels-Associated Diseases (MiNICAD), Universidad de Talca, Talca, Chile

*For correspondence:
decher@staff.uni-marburg.de

[†]These authors contributed equally to this work

Competing interests: The authors declare that no competing interests exist.

**Abstract** Two-pore-domain potassium (K₂ₚ) channels are key regulators of many physiological and pathophysiological processes and thus emerged as promising drug targets. As for other potassium channels, there is a lack of selective blockers, since drugs preferentially bind to a conserved binding site located in the central cavity. Thus, there is a high medical need to identify novel drug-binding sites outside the conserved lipophilic central cavity and to identify new allosteric mechanisms of channel inhibition. Here, we identified a novel binding site and allosteric inhibition mechanism, disrupting the recently proposed K⁺-flux gating mechanism of K₂ₚ channels, which results in an unusual voltage-dependent block of leak channels belonging to the TASK subfamily. The new binding site and allosteric mechanism of inhibition provide structural and mechanistic insights into the gating of TASK channels and the basis for the drug design of a new class of potent blockers targeting specific types of K₂ₚ channels.
DOI: https://doi.org/10.7554/eLife.39476.001

## Introduction

Two-pore-domain potassium (K₂ₚ) channels are characterized by a unique structure and pharmacology. Crystallographic K₂ₚ channel structures revealed that the extracellular M1-P1 linkers of the dimeric channels form a large extracellular 'cap' (*Brohawn et al., 2012*; *Miller and Long, 2012*). TWIK-related acid-sensitive K⁺ (TASK) channels sense pH-variations in the physiological range and are modulated by volatile and intravenous anesthetics. TASK-like currents are found in various tissues and TASK channels are drug targets for many diseases, including diabetes (*Pisani et al., 2016*), obstructive sleep apnea (*Kiper et al., 2015*), pulmonary hypertension (*Bohnen et al., 2017*), and arrhythmias, like atrial fibrillation (*Kiper et al., 2015*; *Liang et al., 2014*; *Limberg et al., 2011*) or conduction disorders (*Decher et al., 2011*; *Friedrich et al., 2014*). As in other K⁺ channels, there is a lack of selective K₂ₚ channel blockers, since drugs preferentially bind to a conserved binding site located in the central cavity of potassium channels (*Decher et al., 2004*; *Marzian et al., 2013*). Thus, for the future rational drug design of specific K₂ₚ channel blockers it is crucial to identify binding sites outside the conserved water filled lipophilic central cavity and new allosteric -non pore plugging- mechanisms of channel inhibition like recently described for TREK channel blockers

binding to the unique cap structure (*Luo et al., 2017*). Strikingly, the recently reported $K_{2P}$ channel crystal structures revealed putative binding sites outside the central cavity, the 'side fenestrations', which are openings towards the lipid bilayer of the membrane (*Brohawn et al., 2012*; *Miller and Long, 2012*). Accordingly, it was suggested that drugs and cellular lipids could bind to these locations (*Miller and Long, 2012*). In fact, norfluoxetine, the active metabolite of the antidepressant drug fluoxetine (Prozac), which blocks TREK channels, binds to these lateral fenestrations of TREK-2 (*Dong et al., 2015*). Despite recent advances in the understanding of the molecular pharmacology of TREK channel blockers (*Dong et al., 2015*; *Luo et al., 2017*) and activators (*Lolicato et al., 2017*), little is currently known about allosteric $K_{2P}$ channel block mechanisms.

$K_{2P}$ channels were crystallized in two different states, the so called 'up' and 'down' state (*Brohawn et al., 2014*). The 'up' state is primarily achieved by a lateral movement of the M4 segments, resulting in the closure of the side fenestrations. Brohawn *et al.* proposed that the 'up' state reflects the conductive state of the channels (*Brohawn et al., 2014*). However, the fact that $K_{2P}$ channels were crystallized in both conformations, each with different potassium occupancies in the selectivity filter, suggested that there is not a tight coupling between the movement of the inner helices and the selectivity filter gate and that the channels can open from both conformations. Consistently, recent studies propose a gating model for TREK channels in which the channels can in fact open from the 'down' state, albeit opening preferentially occurs from the 'up' state (*Brennecke and de Groot, 2018*; *McClenaghan et al., 2016*). The gating at the selectivity filter itself occurs in a voltage-dependent manner with potassium ions as the voltage-sensing particles, resulting in increased outward currents upon depolarization (*Schewe et al., 2016*).

In our current study, we found that the local anesthetic bupivacaine blocks TASK-1 in a voltage-dependent manner, which is a very unusual feature for a 'leak' channel blocker. Using functional mutagenesis screens, *in silico* drug dockings and molecular dynamics (MD) simulations, we identified the binding site of this unusual $K_{2P}$ channel blocker, which is located in the side fenestrations of TASK-1, underneath the second pore helices. Strikingly, the drug is positioned so far laterally in the fenestrations that it does not cause a classical pore occlusion. Here we describe a novel allosteric block mechanism disrupting the $K^+$-flux gating at the selectivity filter which results in TASK-1 channels with an unusual inward-rectification and accordingly a voltage-dependent inhibition. The newly identified binding site and allosteric mechanism of $K^+$-flux gating inhibition described here will allow the rational drug design of potent and $K_{2P}$ subfamily specific blockers with specific kinetic features of inhibition.

## Results

### Voltage-dependent block of TASK-1 by impaired outward $K^+$-flux gating

Initially we aimed to investigate the differences between potent TASK-1 blockers such as A1899 or A293 and the less potent local anesthetic bupivacaine. As a first step, we tested whether bupivacaine causes a stereoselective block of TASK-1. Using the *Xenopus* oocyte expression system both bupivacaine enantiomers and the racemic drug blocked TASK-1 outward currents with equal potencies (*Figure 1—figure supplement 1a,b*), similar to that previously described for the racemic drug (*Leonoudakis et al., 1998*) and the rate of inhibition was not significantly different for the enantiomers (*Figure 1—figure supplement 1c*). As there is no enantioselectivity for TASK-1 inhibition, the subsequent experiments were carried out with the racemic drug that is most commonly used. To test for a voltage-dependence of TASK-1 inhibition of the different compounds, we performed voltage-clamp experiments under symmetrical potassium concentrations. Surprisingly, bupivacaine preferentially blocks outward currents (*Figure 1a–c*), converting the linear current-voltage relationship of TASK-1 into an inwardly rectifying channel behavior (*Figure 1a,b*). This effect is caused by a voltage-dependent inhibition of TASK-1 (*Figure 1c,d*), a feature which has never been reported before for a $K_{2P}$ channel blocker. This blocking characteristic was not present for previously described TASK-1 channel blockers, such as A1899 (*Figure 1f–i*) and A293 (*Figure 1k–n*). Moreover, the voltage-dependence of block is only present for TASK-1 and TASK-3 channels (*Figure 1p*), but not for other $K_{2P}$ channels like TREK-1, TREK-2, TRAAK, TRESK or TASK-2 (*Figure 1p* and *Figure 1—figure supplement 2*). These data argue against a sole contribution of the protonation state and that only

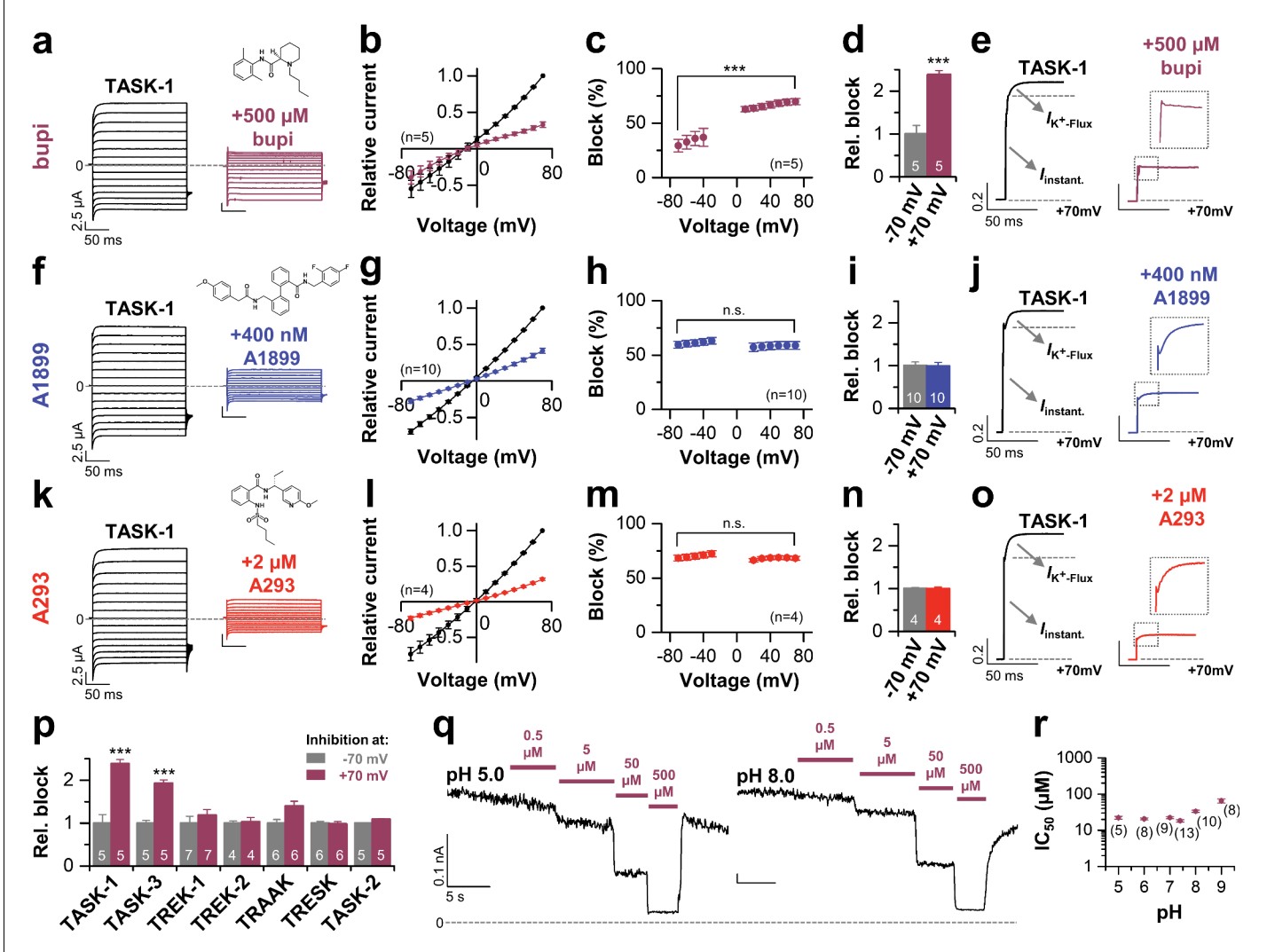

**Figure 1.** Voltage-dependent block of TASK-1 by impaired outward K$^+$-flux gating. (a) Representative current traces of TASK-1 injected oocytes before (black) and after (purple) application of 500 μM bupivacaine, recorded in a symmetrical potassium solution (KD96) at potentials ranging from −70 mV to +70 mV. (b) Average IV-relationships recorded before and after drug application. (c) Analyses of the inhibition by bupivacaine at different voltages. (d) Relative block by bupivacaine comparing the inhibition at −70 and +70 mV, recorded under symmetrical potassium concentrations. (e) Mean current kinetics before (left) and after (right) application of 500 μM bupivacaine, illustrating the instantaneous ($I_{instant}$) and K$^+$-flux gated ($I_{K^+\text{-flux}}$) current component of the TASK current. Note, the lack of a K$^+$-flux gated component after application of bupivacaine. (f–j) As in (a–e), but for A1899 or (k–o) for A293. (p) Relative block by bupivacaine comparing the inhibition of different K$_{2P}$ channels at −70 mV and +70 mV, recorded under symmetrical potassium concentrations. (q) Representative measurements of TASK-3 channels from inside-out macropatches showing the inhibition by increasing concentrations of bupivacaine at pH 5.0 (left) and pH 8.0 (right), recorded at a constant potential of +60 mV. (r) IC$_{50}$ values of TASK-3 inhibition by bupivacaine at various intracellular pH levels. Data are represented as mean ± S.E.M.. The numbers of experiments (n) are indicated within brackets or as small insets in the respective bars.

DOI: https://doi.org/10.7554/eLife.39476.002

The following figure supplements are available for figure 1:

**Figure supplement 1.** Block of TASK-1 channels by different bupivacaine enantiomers.
DOI: https://doi.org/10.7554/eLife.39476.003

**Figure supplement 2.** Block of different K$_{2P}$ channels by bupivacaine.
DOI: https://doi.org/10.7554/eLife.39476.004

**Figure supplement 3.** Block of K$^+$-flux gating by bupivacaine.
DOI: https://doi.org/10.7554/eLife.39476.005

**Figure supplement 4.** TASK-1 block by bupivacaine does not depend on the extracellular potassium concentration.
DOI: https://doi.org/10.7554/eLife.39476.006

charged molecules cause the voltage-dependent inhibition. In addition, varying the intracellular pH in inside-out patches did not alter the TASK-3 sensitivity for bupivacaine, which further supports that the voltage-dependence of TASK-1 and TASK-3 inhibition is most likely not caused by charged bupivacaine molecules (*Figure 1q,r*).

TASK-1 channels conduct with an instantaneous current component and a time-dependent activating component, most likely reflecting increased currents due to the $K^+$-flux gating process at the selectivity filter (*Figure 1e*). Most importantly, in addition to the preferential block of outward currents by bupivacaine, the drug also significantly abrogates the time-dependent $K^+$-flux-gated outward current component of TASK-1 (*Figure 1e* and *Figure 1—figure supplement 3*). This kinetic of inhibition of the outward currents is very unusual for $K_{2P}$ channel blockers, which normally just cause a 'shrinking' of the overall current amplitude without affecting channel kinetics (*Putzke et al., 2007*; *Streit et al., 2011*). Accordingly, this effect was not present for the TASK-1 blockers A1899 or A293 (*Figure 1j,o* and *Figure 1—figure supplement 3*). We have previously reported that TASK-1 inhibition by the open channel blocker A1899 is attenuated by extracellular potassium (*Streit et al., 2011*). In contrast, while increasing extracellular potassium concentration gradually decreased TASK-1 block by A1899, the inhibition of TASK-1 by bupivacaine was not affected (*Figure 1—figure supplement 4*). Summarizing, our data indicate that bupivacaine does not act as a classical open channel blocker and is utilizing a novel $K_{2P}$ channel inhibiting mechanism impairing voltage-dependent $K^+$-flux gating. Thus, the binding site of bupivacaine is most likely different to that of TASK-1 open channel blockers, such as A1899.

## Bupivacaine utilizes a novel binding site in TASK-1

We have previously described the drug-binding site for A1899, the first highly potent blocker of TASK-1 channels. A1899 binds from the intracellular side to the central cavity of TASK-1 by interacting with several residues of the pore lining M2 and M4 segments and residues of the $K^+$ channel signature sequence near the selectivity filter, ultimately resulting in a pore occlusion. In order to map the binding site of bupivacaine in TASK-1, we first performed a systematic alanine mutagenesis screen of the M2 and M4 segments and the threonine residues of the P1 and P2 loops, determining the inhibition by bupivacaine in the oocyte expression system (*Figure 2*). We identified several residues for which the bupivacaine sensitivity was diminished by mutations at the binding site that was previously identified for A1899, including I118 in M2 and I235, G236, L239 and N240 in M4 (classic 'hits') (*Figure 2a–c*). However, we also identified novel 'hits' including C110A, M111A, A114V, Q126A and S127A in the M2 segment, as well as V234A and F238A in the M4 segment (*Figure 2a–c*). In the pore signature sequence, we identified T198 of the second pore loop (P2) to be relevant for drug block, while the homologous mutation (T92A) in the first pore loop did not affect bupivacaine sensitivity (*Figure 2b,c*). Next, we used a TASK-1 homology model based on the down state of TRAAK to make further predictions and to illustrate the residues of the bupivacaine binding site (*Figure 2d*) which is clearly different to that of A1899 (*Figure 2e*). The bupivacaine binding site is more 'V'-like shaped, as the compound appears to interact with residues that are in the upper sideward part of the cavity, such as C110, M111 and A114 (*Figure 2d*). These data suggest that bupivacaine might be located laterally underneath the pore helices, in the side fenestrations that were previously described for other $K_{2P}$ channels. Therefore we alanine scanned the M1-P1 and the M3-P2 pore helices and the M3 segment, which are, according to the crystal structures of other $K_{2P}$ channels, located at the outer entry to the lateral fenestrations. Here we identified a strong reduction of the bupivacaine sensitivity by mutations of the second threonines of the pore signature sequence (T93C, T199C) in P1 and P2 (*Figure 2f*), residues previously described to alter the $K^+$-flux gating of $K_{2P}$ channels (*Schewe et al., 2016*). Given the two-fold rhomboid symmetry of $K_{2P}$ channels, only the second pore helices are expected to be located above the side fenestrations (*Figure 3—figure supplement 1*). Strikingly, we only identified strong sensitivity changes for mutations located in the P2 pore helix (F194A, T198C) (*Figure 2f*). In addition, we also observed weak but significant reduction of the bupivacaine sensitivity for the L171V mutant in the M3 segment (*Figure 2f*), which is expected to be located at the entry to the lateral fenestrations.

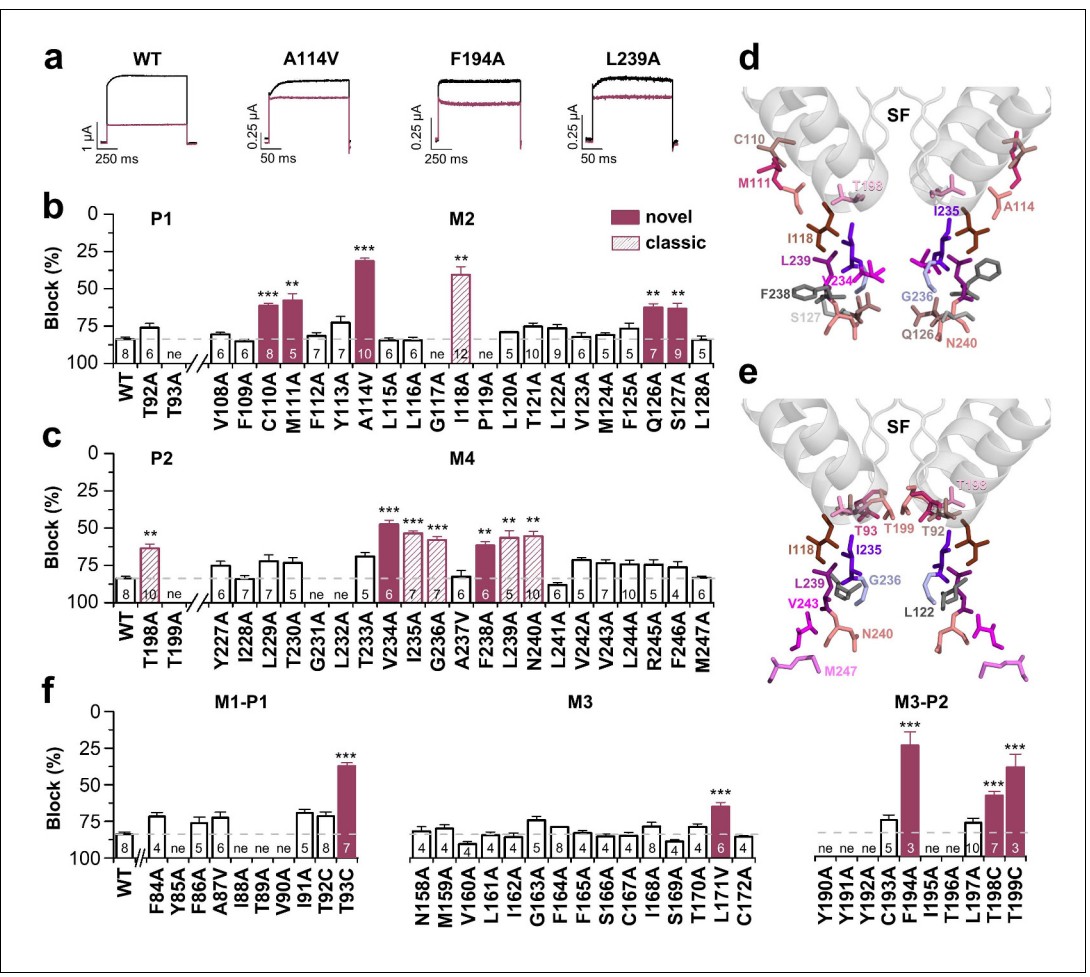

**Figure 2.** Bupivacaine utilizes a novel binding site in TASK-1. (a) Representative current traces of wild-type TASK-1 (WT) and mutant channels expressed in oocytes before (black) and after (purple) application of 500 µM bupivacaine. (b) Percentage of inhibition by 500 µM bupivacaine of wild-type TASK-1 and alanine mutants of the P1 signature sequence, the M2 segment and (c) the P2 signature sequence and the M4 segment. Residues identified as drug-binding sites ('hits'), which were previously also reported for the binding site of the open channel blocker A1899 ('classical'), are illustrated as striated bars and the 'hits' of the novel bupivacaine binding site are depicted as filled bars ('novel'). (d) Amino acids involved in binding of bupivacaine and (e) the respective A1899 binding mode, according to *Ramírez et al. (2017)*, both illustrated in a TRAAK-OO (PDB ID: 3UM7) based TASK-1 homology model. SF indicates the selectivity filter. (f) Analyses of the percentage of inhibition by 500 µM bupivacaine for TASK-1 alanine mutants of the M1-P1, M3 and M3-P2 segments. ne, not expressed. Data are represented as mean ± S.E.M.. The numbers of experiments (n) are indicated within the respective bars.
DOI: https://doi.org/10.7554/eLife.39476.007

## TASK-1 channels are fenestrated providing the bupivacaine binding site

As there is no crystal structure of TASK-1 reported so far, we generated different TASK-1 homology models based on the available crystal structures, namely TREK-2 with both fenestrations closed (TREK-2-CC), TRAAK with both (TRAAK-OO) and one fenestration open (TRAAK-CO), TREK-2 with both fenestrations opened (TREK-2-OO), as well as TWIK-1 that has two wide side fenestrations (TWIK-1-OO). Next, we docked bupivacaine into the different models and analyzed the docking solutions towards the interaction with the residues we identified in our alanine mutagenesis screen. Here we noted that the best overlap with our functional data is given for docking solutions with bupivacaine located in the side fenestrations of the different TASK-1 homology models. Here, we noted that with gradually increasing the diameter of the side fenestration (*Figure 3a–e*) the number of bupivacaine docking solutions (conformers) in the fenestrations and the number of clusters in the

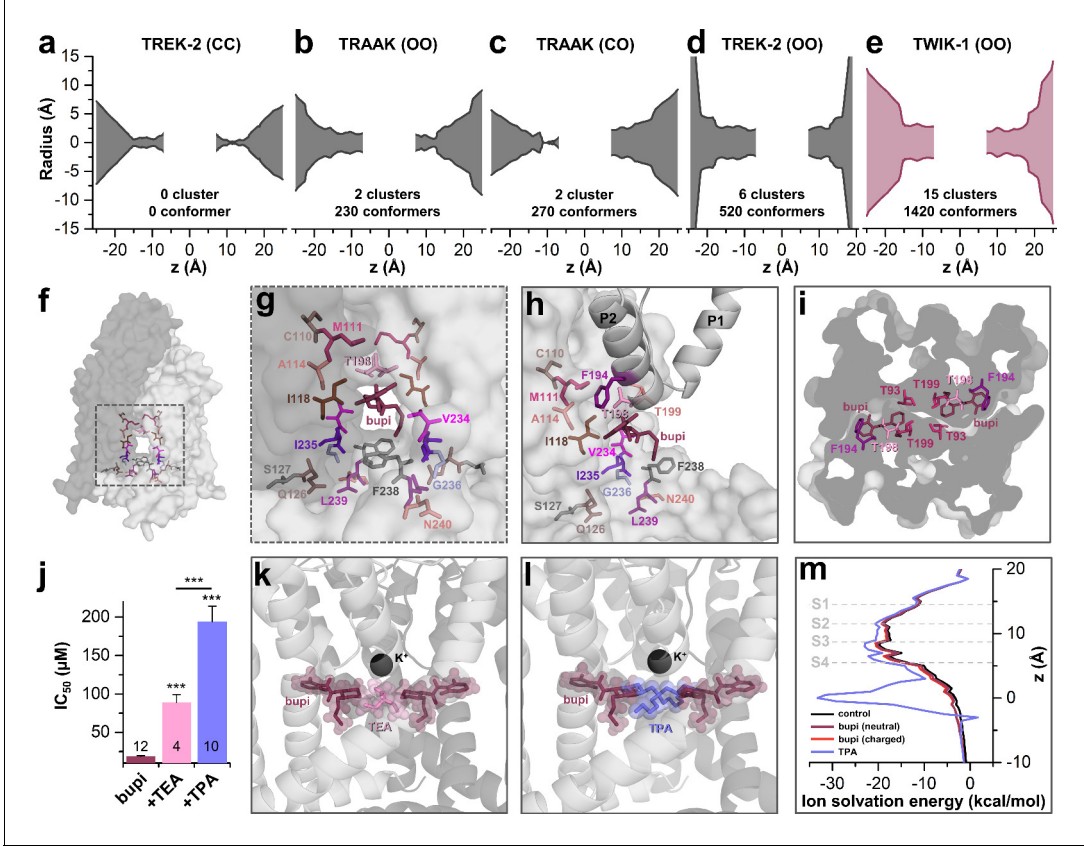

**Figure 3.** TASK-1 channels are fenestrated providing the bupivacaine binding site. (**a–e**) Hole analyses illustrating the diameter of the side fenestrations in different TASK-1 homology models together with the number of docking solutions (conformers) located in the fenestrations and the number of significant clusters formed by the different conformers. 'C' indicates a closed and 'O' an open fenestration. The TASK-1 homology models were based (**a**) on TREK-2-CC (PDB ID: 4BW5), (**b**) TRAAK-OO (PDB ID: 3UM7), (**c**) TRAAK-CO (PDB ID: 4I9W), (**d**) TREK-2-OO (PDB ID: 4XDK), and (**e**) TWIK-1-OO (PDB ID: 3UMK). (**f**) TASK-1 homology model based on TWIK-1 illustrating the amino acid 'hits' in the M2 and M4 segments identified by alanine scanning. Note that the 'hits' primarily line the side fenestrations. (**g**) Enlargement of (**f**) with bupivacaine located in the side fenestrations illustrating the involvement of the M2 and M4 hits. (**h**) View showing the localization of bupivacaine under the second pore helix together with the respective M2 and M4 'hits'. (**i**) Cross section view from the bottom of the TASK-1 model visualizing the two side fenestrations and the localization of bupivacaine together with the 'hit' residues of the pore helices. (**j**) $IC_{50}$ of bupivacaine on TASK-3 channels recorded in inside-out macropatch clamp experiments alone or in the presence of TEA or TPA. (**k**) Binding mode of bupivacaine together with that of TEA and (**l**) TPA, respectively. (**m**) Ion solvation free energy profile for potassium ions in the TWIK-1 based TASK-1 pore homology model alone (control), in the presence of neutral or charged bupivacaine or TPA.

DOI: https://doi.org/10.7554/eLife.39476.008

The following figure supplements are available for figure 3:

**Figure supplement 1.** Positioning of bupivacaine in the side fenestrations underneath the second pore helices.

DOI: https://doi.org/10.7554/eLife.39476.009

**Figure supplement 2.** Conserved binding site and block of TASK-1 and TASK-3 channels by bupivacaine.

DOI: https://doi.org/10.7554/eLife.39476.010

fenestrations strongly increased (*Figure 3a–e*). Genes encoding the crystallized TWIK and TREK/TRAAK channels are paralogs of *KCNK3* encoding TASK-1 and within these channels TWIK-1 is the closest relative to TASK-1 with a sequence identity of 26.1%. These genetic lines of evidence are further supported by the fact that TWIK-1 and TASK-1 share functional similarities, that is they change their permeability upon extracellular acidification and they form heterodimeric channels (*Ma et al., 2012*; *Plant et al., 2012*). Strikingly, the TWIK-1 based homology model showed by far the most significant clusters (15) and conformers (1420) in the side fenestrations (*Figure 3e*). In addition, analyzing the free energy of the docking solutions (MM-GBSA) (*Ramírez et al., 2017*) that fit the best to our functional data, revealed that the most stable docking solutions are generated using the TWIK-based TASK-1 homology model. Thus, we propose that TASK-1 channels are likely to exhibit large

side fenestrations similar as other $K_{2P}$ channels. Analyzing the TWIK-1 based TASK-1 homology model that lead to the best docking solutions, the residues we identified in the alanine scan line the side fenestrations (*Figure 3f*). In the best docking solution, bupivacaine is located in the side fenestrations (*Figure 3g*) located underneath the M3-P2 pore helices (*Figure 3h* and *Figure 3—figure supplement 1*). The 'classical' drug-binding site residues I118 of the M2 segment and I235 of the M4 segment have an intermediate location between the central cavity and the side fenestrations and also interact with bupivacaine (*Figure 3g–h*). Remarkably, the 'atypical' and novel residues identified, with C110, M111 and A114 of M2, F194, T198 and T199 of the M3-P2 linker and V234 and F238 of M4, face towards the side fenestrations and the drug (*Figure 3g,h*). The drug is located in a thin sewer connecting the side fenestration with the central cavity, explaining the preferential interaction with the second of the two pore helices, as this is positioned directly above the drug (*Figure 3i* and *Figure 3—figure supplement 1*).

## Bupivacaine causes an allosteric TASK channel inhibition

TASK-3 channels are blocked by bupivacaine with the similar mechanism (*Figure 1p*) and sensitivity (*Figure 3—figure supplement 2*) as TASK-1 channels, which is in line with our observation that the binding site is fully conserved between these two closely related channels (*Figure 3—figure supplement 2a*). As TASK-3 channels show a very pronounced functional expression (*Rinné et al., 2015*), we performed inside-out macropatch clamp experiments using this channel. Here we probed for a competition of bupivacaine, which is located in the side fenestrations, with the pore blocking quaternary ammonium compounds TEA and TPA (*Piechotta et al., 2011*). Consistent with the binding site of bupivacaine we have proposed here, the smaller TEA causes a less pronounced competition and increase in $IC_{50}$ than TPA, as the binding sites only partially overlap (*Figure 3j,k*), while TPA is larger and protrudes up to the entry of the side fenestrations (*Figure 3j,l*). This data is in line with our final docking solution of bupivacaine located in the side fenestrations, at a position that does not allow a physical occlusion of the selectivity filter or prevention of the passage of potassium ions through the inner cavity. Quaternary ammonium compounds do however, in addition to a spatial pore plugging mechanism, also disturb the ion passage by an electrostatic interaction with the permeating ion (*Faraldo-Gómez et al., 2007*; *Kutluay et al., 2005*). Here we calculated the ion solvation free energy profiles which show that TPA induces an ion solvation free energy barrier due to its charge, an effect that was however not present for neutral and charged bupivacaine (*Figure 3m*). These data indicate that bupivacaine does not disrupt $K^+$ conductance due to an electrostatic repulsion effect. In addition, it supports our hypothesis that it is not the charged bupivacaine causing the voltage-dependence of block and that an allosteric inhibition mechanism causes the untypical $K_{2P}$ channel inhibition.

## MD simulations - binding in the fenestrations prevents fenestration closure

Next, we performed 100 ns of MD simulations with bupivacaine located in the binding site of the side fenestration, analyzing the number of contacts with the key residues identified in our alanine scanning approach. These simulations revealed a stable binding of bupivacaine to the side fenestration and confirmed almost all interactions with the residues that we identified in our functional screen. Only for the residues C110 in the upper and S127 in the lower M2 segment, we observed no contacts with the drug during the MD simulations, as these were not perfectly facing towards the bupivacaine-binding site. These 'hits' were considered as false positives that might have allosterically influenced drug binding or inhibition, as they are in close proximity to the compound and yet do not have direct contact. Notably, we found a high number of contacts with F194, T198 and T199 that are located in the second pore helix (*Figure 4a–c*). As T199 determines the ion occupancy of the selectivity filter in $K_{2P}$ channels (*Schewe et al., 2016*), an interaction with this residue and the second pore helix might allosterically interfere with $K^+$ occupancy and conductance of the selectivity filter. In addition, we also observed contacts during the MD simulations with residues of the M3 segment, including S166, C167, T170 and L171 (*Figure 4b,c*), which is consistent with the significant changes in sensitivity we observed for the M3 mutations C167D, L171V and the S116A/C167A/T170A triple mutation (*Figure 2f* and *Figure 4—figure supplement 1*). The data further support that bupivacaine is not occluding the central cavity and is located in the side fenestrations, interacting with residues

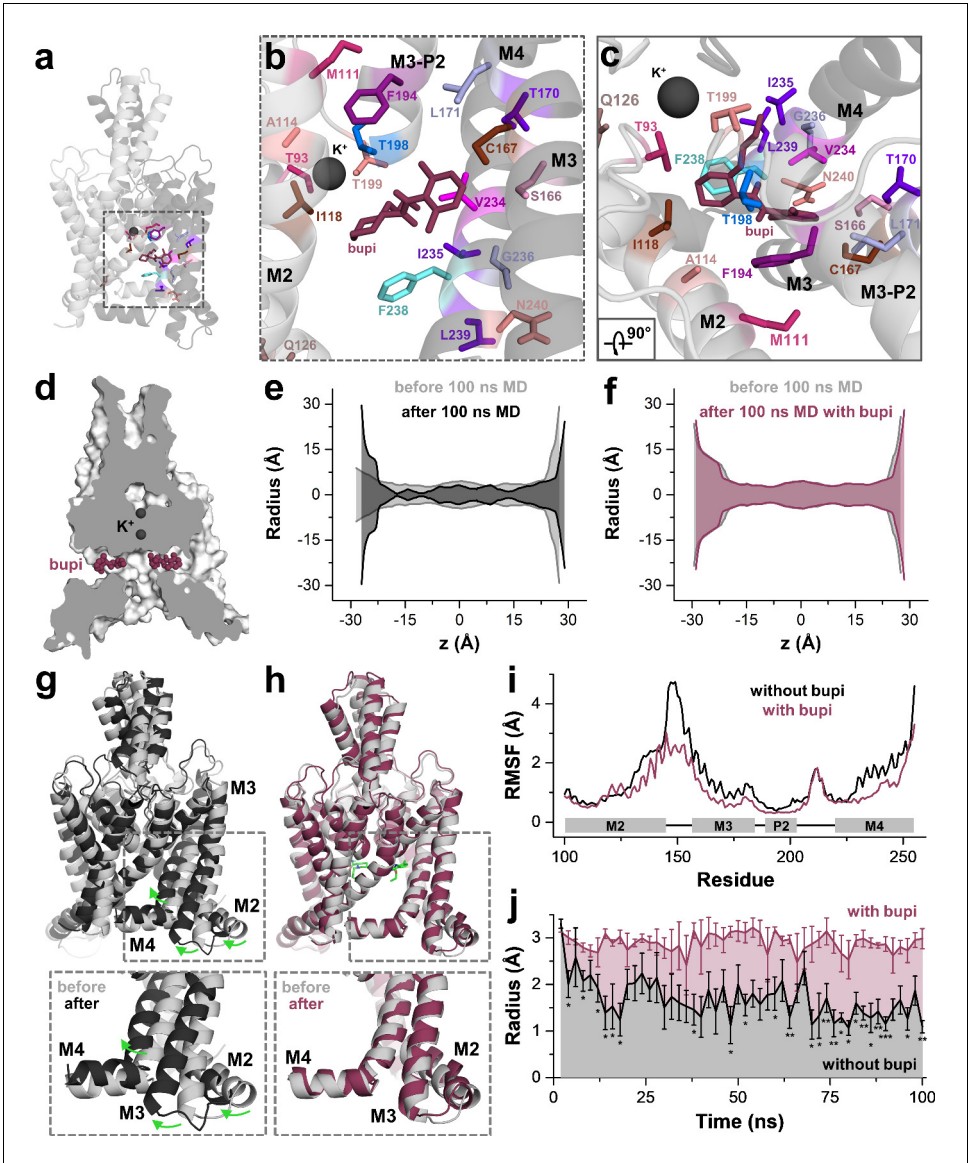

**Figure 4.** MD simulations - binding in the fenestrations prevents fenestration gating. (a) Side view on a TWIK-1 based TASK-1 homology model with bupivacaine located in the side fenestrations after 100 ns MD simulations. (b–c) Zoom-ins illustrating 'hit' residues of the alanine scan, which were confirmed in the MD simulations to have contacts with bupivacaine. (d) TASK-1 homology model based on TWIK-1 depicting bupivacaine within the two side fenestrations. (e) Hole analysis before (light gray) and after (gray) 100 ns MD simulations revealing a collapse of the side fenestrations when the channels move to the 'up' state. (f) Hole analysis as in (e), but in the presence of bupivacaine (purple), which prevents the movement to the 'up' state and the concomitant collapse of the fenestrations. (g) TASK-1 homology model in the absence of bupivacaine before (gray) and after 100 ns of MD simulations (black). Green arrows indicate the movement of the M2, M3 and M4 segments causing the 'down' to 'up' transition. (h) TASK-1 homology model in the presence of bupivacaine (green) before (gray) and after (purple) 100 ns, illustrating the lack of the M2, M3 and M4 movements. (i) Root-mean-square fluctuations (RMSF) calculated for the alpha carbons of the M2, M3, P2 and M4 segments in the absence and presence of bupivacaine. (j) Quantitative analysis of the bottleneck radius of the side fenestrations over the time course of 100 ns MD simulations (n = 4) in the absence (gray) and presence (purple) of bupivacaine. Data are represented as mean ± S. E.M.

DOI: https://doi.org/10.7554/eLife.39476.011

The following figure supplements are available for figure 4:

**Figure supplement 1.** Reduced bupivacaine inhibition by M3 mutants.

DOI: https://doi.org/10.7554/eLife.39476.012

*Figure 4 continued*

**Figure supplement 2.** Induced fit docking experiments predict a bupivacaine binding mode highly similar to that of classical docking experiments.
DOI: https://doi.org/10.7554/eLife.39476.013

**Figure supplement 3.** MD simulations - binding in the fenestrations prevents fenestration gating.
DOI: https://doi.org/10.7554/eLife.39476.014

of the second pore helix and that bupivacaine is positioned so far laterally that it is able to form weak contacts with residues of the M3 segment.

Next, we used a complementary approach utilizing induced fit docking experiments with four different TASK-1 homology models (Material and methods and *Figure 4—figure supplement 2*). Here, the highest success rate of the induced fit dockings were again obtained in a TASK-1 model based on TWIK-1-OO (*Figure 4—figure supplement 2a*). Given the highest success rate in the TWIK-1-based homology model and that TWIK-1 is the closest relative of the crystallized K_{2P} channels, we selected the pose with the lowest IFD energy that is, as described above, also consistent with the TEA versus TPA competition experiments. The induced fit docking solution in the TWIK-OO based TASK-1 homology model is very similar to the one obtained by our classical docking approach (*Figure 4—figure supplement 2b–c*), confirming the binding site we have proposed above.

Similar as previously described for TWIK-1 (*Jorgensen et al., 2016*), TASK-1 channels converted into the 'up' state during MD simulations (100 ns), resulting in a closure of the side fenestrations (*Figure 4e,j*). Strikingly, this conversion to the 'up' state was prevented during four replicates of 100 ns MD simulations, when charged or uncharged bupivacaine was located in the binding site of the fenestrations (*Figure 4d,f,j* and *Figure 4—figure supplement 3*). This transition of the channel from the 'down' to 'up' state, which exclusively occurs in the absence of bupivacaine, is caused by a coordinated movement of the M2, M3 and M4 segments (*Figure 4g,h* and *Video 1*), similar as previously reported for 100 ns MD simulations of TREK-2 channels (*Dong et al., 2015*). These movements in the absence of bupivacaine are also reflected by an analysis of the root-mean-square fluctuations (RMSF) (*Figure 4i*), revealing that the M2-M3 linker and the M4 segment are much more flexible during MD simulations in the absence of bupivacaine. As these movements are expected to concomitantly close the side fenestrations, we have quantitatively analyzed this transition, monitoring the bottleneck diameter of the fenestration in MD simulations over time (*Figure 4j*). Here the bottleneck radius of the fenestrations quickly and significantly decreases in the absence of bupivacaine, while bupivacaine stabilizes the bottleneck radius of the side fenestrations to remain at about 3 Å over the whole time period of the 100 ns MD simulations (*Figure 4j*). Thus, bupivacaine prevents closure of the side fenestrations and the conversion of the channel to the 'up' state. However, whether TASK-1 channels gate in a similar way as TREK-1 channels and whether they physiologically convert into an 'up'-state-like conformation currently remains unknown. Thus, it also remains an open question whether effects of bupivacaine on the architecture of the fenestrations contribute to the allosteric mechanism of inhibition.

## Bupivacaine stabilizes the closed state of TASK channels

Next, we performed inside-out single channel measurements of TASK-3 to support the inhibition mechanism proposed above. TASK-3 has longer open times and a higher single channel conductance than TASK-1 and thus is more suitable for an analyses of the inhibition mechanism on a single channel level. Similar as in the whole-cell recordings, bupivacaine caused a preferential inhibition of outwardly-directed single channel events when applied form the intracellular side of the membrane (*Figure 5a*). In a

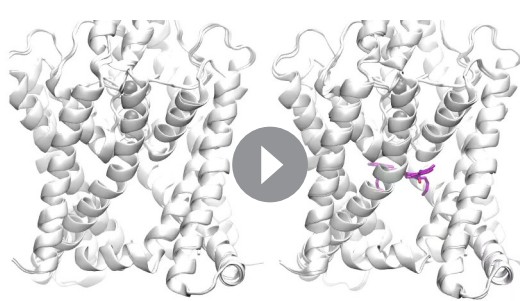

**Video 1.** MD simulation of the TASK-1 homology model with and without bupivacaine. TASK-1 homology model in the absence (left) and the presence (right) of bupivacaine during 100 ns of MD simulations. Note the lack of major M2, M3 and M4 re-arrangements in the presence of bupivacaine (right).
DOI: https://doi.org/10.7554/eLife.39476.015

concentration that reduced the open probability by approximately fifty percent (*Figure 5c*), bupivacaine did not alter the single channel conductance (*Figure 5b*) and did not shorten the mean open time (*Figure 5d,e*) or burst duration (*Figure 5—figure supplement 1*), effects that are typically observed for open channel blockers. In contrast, bupivacaine stabilized the closed state of the channel resulting in longer closed times (*Figure 5f,g*). In contrast, TEA, which is a classical voltage-dependent blocker of Kv channels and an open channel blocker of $K_{2P}$ channels (*Piechotta et al., 2011*), reduced the open times of TASK-3 (*Figure 5h–j*) and also caused a mild reduction in conductivity of the channels (*Figure 5k,l*). Thus, the mechanism of bupivacaine inhibition is clearly different to that of a classical open channel blocker, as the drug causes an isolated stabilization of the closed state, consistent with a prevention of voltage-dependent $K^+$-flux gating, which would normally allow a higher conductivity at depolarized membrane potentials.

## A novel allosteric inhibition mechanism interferes with the voltage-dependent activation of TASK channels

The binding site and inhibition mechanism of bupivacaine is different to that of pore plugging compounds. Bupivacaine is located laterally in the side fenestrations of TASK-1 channels and interacts with residues of the pore helix, the M2, M3 and M4 segments, while pore blockers such as A1899

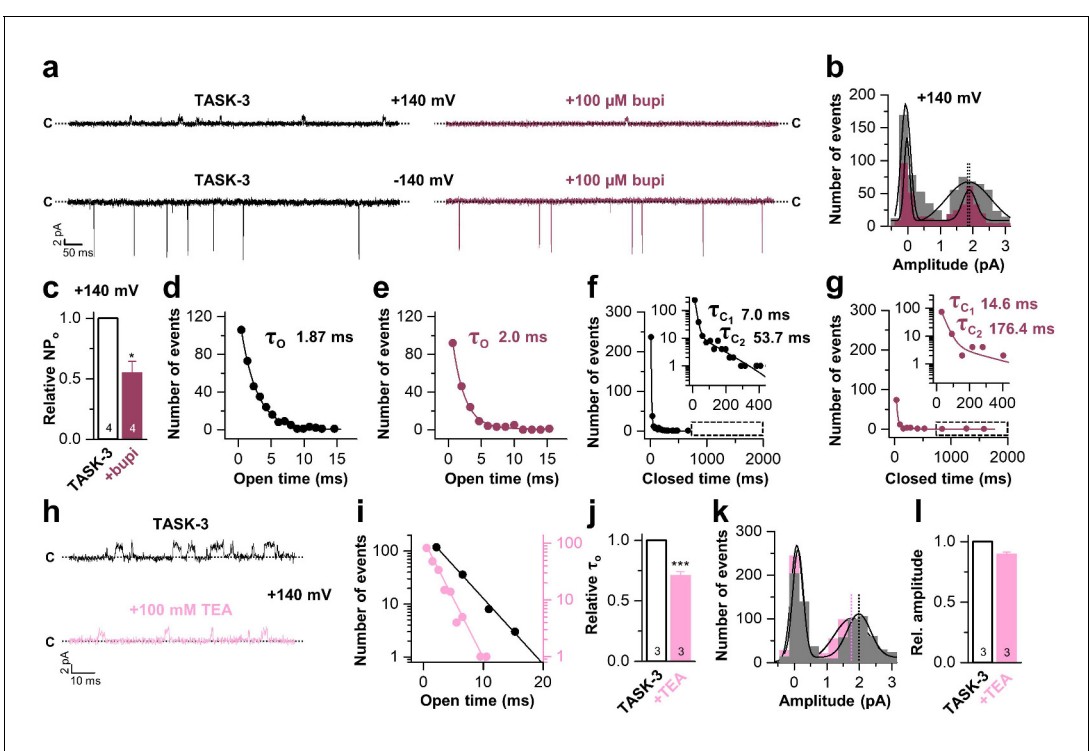

**Figure 5.** Bupivacaine stabilizes the closed state of TASK channels. (**a**) Representative inside-out single channel recordings of TASK-3 at +140 and −140 mV before (left, black) and after (right, purple) application of 100 µM bupivacaine and (**b**) amplitude histogram of the events recorded at +140 mV (n = 4). (**c**) Change in relative open probability ($NP_o$) by 100 µM bupivacaine. (**d**) Graphical illustration of the open times before and (**e**) after drug application (n = 4). (**f**) Analyses of the closed time before and (**g**) after application of 100 µM bupivacaine (n = 4). (**h**) Representative inside-out single channel recordings of TASK-3 at +140 mV before (up, black) and after (down, light pink) application of 100 mM TEA and (**i**) representative illustration of the open times of a TASK-3 containing patch before and after drug application. (**j**) Relative change in open times by 100 mM TEA (n = 3). (**k**) Amplitude histogram of the events recorded at +140 mV (n = 3) and (**l**) the respective analyses of the relative change in single channel amplitude before (black) and after (light pink) application of 100 mM TEA. Data are represented as mean ± S.E.M.. The numbers of experiments (n) are indicated within the respective bars.

DOI: https://doi.org/10.7554/eLife.39476.016

The following figure supplement is available for figure 5:

**Figure supplement 1.** Analyses of the single channel burst characteristics of TASK-3 after application of bupivacaine.

DOI: https://doi.org/10.7554/eLife.39476.017

are located in the central cavity to prevent K$^+$ permeation (*Figure 6a,b*). As bupivacaine located in the side fenestration interferes with the voltage-dependent K$^+$-flux gating, the increased outward currents that are physiologically present upon depolarization are impaired, resulting in a lack of outward rectification and a preferential inhibition of outward currents (*Figure 6c*). In conclusion, the weak voltage-dependent inhibition of TASK channels that we describe here, results from a new allosteric inhibition mechanism, which operates by preventing the voltage-dependent K$^+$-flux gating.

## Discussion

Utilizing systematic alanine scanning mutagenesis, *in silico* docking experiments and MD simulations, combined with voltage-clamp and single channel measurements, we identified the molecular basis for a novel inhibition mechanism for K$_{2P}$ channels, which functions in an allosteric and voltage-dependent manner. While for TASK channels no crystallographic structures have been reported so far, the data describing the bupivacaine binding site, together with the allosteric inhibition mechanism we present here, allows valuable predictions regarding the structure of these channels. We propose that TASK-1 channels must have lateral fenestrations large enough to accommodate drugs and that binding of bupivacaine in the fenestrations allosterically prevents the voltage-dependent K$^+$ flux gating.

While we identified specific residues to exclusively affect block by bupivacaine and not A1899, there is still a partial overlap to the binding site of the open channel blocker A1899 (i.e. residues I118, I235, G236, L239 and N240). This apparent discrepancy can be explained by the fact that many of the A1899 binding residues (*Streit et al., 2011*) are not strictly 'pore facing' and are likely to be also accessible from the side fenestrations (*Ramírez et al., 2017*). In general, one has to consider that reduced inhibition of a mutant does not necessarily imply binding at the respective site, as it is

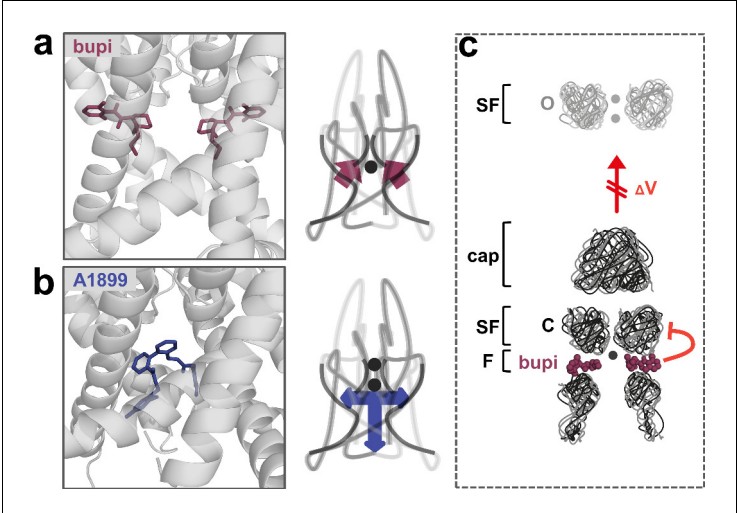

**Figure 6.** A novel allosteric inhibition mechanism interferes with the voltage-dependent activation of TASK channels. (a) Illustration of the bupivacaine binding mode in the side fenestrations (left), preventing K$^+$-flux gating, which requires a K$^+$ filled selectivity filter (right cartoon). The closed state or collapsed selectivity filter is indicated by one potassium ion (black dot). (b) A1899 located in the central cavity (left) is binding in an 'anchor'-shaped like structure which occludes the pore and thus prevents K$^+$ permeation (right cartoon). Here the K$^+$ occupancy of the selectivity filter is not disturbed (illustrated by two black dots in the selectivity filter). (c) Schematic illustration of the proposed gating model of TASK-1. The upper panel illustrates the selectivity filter in the open state and the lower panel the TASK channel. 'C' indicates the closed state conformation of the selectivity filter (one potassium ion located before the collapsed filter) and 'O' indicates the open state with a K$^+$ filled selectivity filter (here two potassium ions illustrated). SF indicates the selectivity filter and F the side fenestrations. The voltage-dependent (ΔV) opening of the selectivity filter gate (K$^+$-flux gating) preferentially results in the typical outwardly rectifying K$^+$ currents. The presence of bupivacaine in the side fenestrations prevents the voltage-dependent K$^+$-flux gating, resulting in reduced outward currents at depolarized potentials and thus a voltage-dependent inhibition.
DOI: https://doi.org/10.7554/eLife.39476.018

conceivable that the mutation affects the block allosterically. This caveat has to be specifically considered when studying mutations that might simultaneously interfere with drug binding and the gating mechanism, like in the case of T199C (*Schewe et al., 2016*). However, the fact that we conclusively mapped a binding site that includes a large set of residues and that this site was confirmed by *in silico* dockings, MD simulations and TEA/TPA competition experiments, must lead to the conclusion that we have identified the most likely binding mode for bupivacaine.

TASK channels provide two side fenestrations as putative allosteric binding sites for the most likely channel inhibition we describe here. The slope factor of the bupivacaine dose-response curve does not indicate a cooperativity of inhibition, suggesting that binding in one of the fenestrations does not facilitate the binding in the second. Thus, it is possible that binding to one of the fenestrations is sufficient to cause channel inhibition or that both sites are occupied by the drug, but channel inhibition is achieved in an additive and non-cooperative manner.

Bupivacaine located in the side fenestrations interacts with residues of the second pore helix. Noteworthy, mutations homologous to T199C of P2 were reported to alter $K^+$-flux gating of $K_{2P}$ channels by changing the ion occupancy in the selectivity filter (*Schewe et al., 2016*). Thus, bupivacaine interacting with residues of the second pore helix (F194, T198 and T199) might interfere with the $K^+$-flux gating by disturbing the initiation of the proper $K^+$ occupancy in the selectivity filter. In this context, it might be possible that the ability of bupivacaine to selectively induce a voltage-dependent inhibition of TASK-1 and TASK-3 reflects that channels of this $K_{2P}$ channel subfamily seem to undergo a distinct gating mechanism, which is disrupted by the drug. This is supported by the fact that TASK-1/-3 channels do not show the typical $Rb^+$-dependent gating, with strongly increased conductivity by $Rb^+$, as the other $K^+$-flux gated $K_{2P}$ channels (*Schewe et al., 2016*).

The local anesthetic bupivacaine is a low potent multi-channel blocking drug, affecting sodium and potassium channels. However, utilizing this compound we were able to identify a novel drug-binding site and allosteric inhibition mechanism, providing the molecular basis for the rational drug design of potent $K_{2P}$ channel modulators with distinct biophysical features. The binding site in the side fenestration is fully conserved for TASK-1 and TASK-3, while it is distinct to that of other $K_{2P}$ channels. Thus, the fenestrations provide the molecular basis for novel $K_{2P}$ channel subtype specific blockers and a new principle of action that might be beneficial for future drugs targeting diseases of excitable tissue, like in arrhythmias, epilepsies and diabetes.

## Materials and methods

### Oocyte preparation, cRNA synthesis and cRNA injection

Oocytes were obtained from anesthetized *Xenopus laevis* frogs as described previously (*Streit et al., 2011*) and incubated in OR2 solution containing (in mM): NaCl 82.5, KCl 2, $MgCl_2$ 1, HEPES 5 (pH 7.5), supplemented with 2 mg/ml collagenase II (Sigma) to remove residual connective tissue. Subsequently, oocytes were stored at 18°C in ND96 solution containing (in mM): NaCl 96, KCl 2, $CaCl_2$ 1.8, $MgCl_2$ 1, HEPES 5 (pH 7.5), supplemented with 50 mg/l gentamycine, 274 mg/l sodium pyruvate and 88 mg/l theophylline. Human TASK-1 channels were subcloned into a pSGEM or pBF1 vector and point mutations were introduced using the QuikChange site-directed mutagenesis kit (Stratagene). Human TASK-3, TREK-2, TRAAK, TRESK, TASK-2 and TREK-1 channel constructs were used in an oocyte expression vector (pSGEM or pBF1). cDNA was linearized with NheI or MluI and cRNA was synthesized with mMESSAGE mMACHINE-Kit (Ambion). The quality of cRNA was tested using gel electrophoresis and cRNA was quantified with the NanoDrop 2000 UV-Vis spectrophotometer (Thermo Scientific). Oocytes were each injected with 50 nl of cRNA.

### TEVC recordings

All TEVC recordings were performed at room temperature (20–22°C) with a TurboTEC 10 CD (npi) amplifier and a Digidata 1200 Series (Axon Instruments) as A/D converter. Micropipettes were made from borosilicate glass capillaries GB 150TF-8P (Science Products) and pulled with a DMZ-Universal Puller (Zeitz). Recording pipettes had a resistance of 0.5–1.2 MΩ and were filled with 3 M KCl solution. ND96 or KD96 were used as recording solutions. KD96 solution contained (in mM): NaCl 2, KCl 96, $CaCl_2$ 1.8, $MgCl_2$ 1, HEPES 5 (pH 7.5). Block was analyzed with voltage steps from a holding potential of −80 mV. A first test pulse to 0 mV of 1 s duration was followed by a repolarizing step to

−80 mV for 1 s, directly followed by another 1 s test pulse to +40 mV. The sweep time interval was 10 s. The IV relationships in ND96 solution were measured from a holding potential of −80 mV (in KD96 from a holding potential of 0 mV) with voltage steps ranging from −70 to +70 mV with 10 mV increments for 200 ms. Data were acquired with Clampex 10 (Molecular Devices), analyzed with Clampfit 10 (Molecular Devices) and Origin 2016 (OriginLab Corporation).

### Inside-out macropatch clamp recordings

Macropatch recordings from *Xenopus laevis* oocytes in inside-out configuration under voltage-clamp conditions were made at room temperature 72–120 hr after injection of 50 nl channel specific cRNA. Thick-walled borosilicate glass pipettes had resistances of 0.3–0.9 MΩ (tip diameter of 5–15 µm) and were filled with extracellular solution containing (in mM): KCl 4, NMDG 116, HEPES 10 and $CaCl_2$ 3.6 (pH was adjusted to 7.4 with KOH/HCl). Bath solution was applied to the cytoplasmic side of excised macropatches via a gravity flow multi-barrel application system and had the following composition (in mM): KCl 120, HEPES 10, EGTA 2 and Pyrophosphate 1. The various appropriate pH levels (pH 5.0, pH 9.0) were adjusted with KOH/HCl. Macroscopic currents from hTASK-3 channels (NM_001282534) were acquired with an EPC10 amplifier (HEKA electronics, Lamprecht, Germany) at a continuous voltage of +60 mV, sampled at 10 kHz and filtered analogue to 3 kHz (−3 dB). Tetra-pentylammonium chloride (TPA-Cl) and bupivacaine hydrochloride monohydrate were purchased from Sigma-Aldrich and stored as stock solutions (100 mM) at −20°C. The drugs were diluted in intracellular bath solution to final concentrations prior to each measurement. For data analysis Igor pro (WaveMetrics Inc, Portland, USA) software was used. The relative steady-state level of blockage for bupivacaine was fitted with the Hill equation: base + (max - base)/[$x_{half}$/x)$^H$], where base = inhibited (zero) current, max = maximum current, x = ligand concentration, $x_{half}$ = value of concentration for half-maximal occupancy of the ligand binding site and H = Hill coefficient.

### Inside-out single channel patch clamp recordings

Single channel patch clamp recordings in the inside-out configuration using *Xenopus* oocytes were performed similar as previously described (*Rinné et al., 2015*). Therefore, the vitelline membranes were manually removed from the oocytes after shrinkage by adding mannitol to the bath solution. All experiments were conducted at room temperature 24–48 hr after cRNA (0.005 ng hTASK-3 per oocyte) injection. Borosilicate glass capillaries GB 150TF-8P (Science Products) were pulled with a DMZ-Universal Puller (Zeitz) and had resistances of 4–6 MΩ. Capillaries were filled with bath solution containing (in mM): KCl 140, HEPES 5, EGTA 1 (pH 7.4 adjusted with KOH/HCl). Bupivacaine and tetraethylammonium chloride (TEA-Cl) were added to the bath solution before each measurement, having final concentrations of 100 µM and 100 mM, respectively. Single channel currents were recorded with an Axopatch 200B amplifier (Axon Instruments), a Digidata 1550B A/D converter (Axon Instruments) and pClamp10 software (Axon Instruments) and were sampled at 15 kHz with the analog filter set to 2 kHz. Voltage pulses were applied from 0 mV (holding potential) to +140 mV and/or −140 mV for 1 s with an interpulse interval of 8 s. Data were analyzed with ClampFit10 and Origin 2016 (OriginLab Corporation).

### Drugs and IC$_{50}$ values

All drugs were dissolved in DMSO, aliquoted, stored at −20°C or room temperature and added to the external solution (ND96 or KD96) just before the recordings. Stock solutions had a concentration of 125 mM (R/S-bupivacaine), 125 mM (R-bupivacaine), 125 mM (S-bupivacaine), 10 mM (A293) and 10 mM (A1899). The IC$_{50}$ was determined from Hill plots using four concentrations for each construct. Final DMSO concentration did not exceed 0.001%.

### Molecular modeling

As previously described (*Ramírez et al., 2017*) five different homology models were generated for human TASK-1 (UniProtKB accession number: O14649) based on TREK-2 with the two fenestrations closed (TREK-2-CC, PDB ID: 4BW5), TREK-2 with both fenestrations opened (TREK-2-OO, PDB ID: 4XDK), TRAAK with two fenestrations opened (TRAAK-OO, PDB ID: 3UM7), TRAAK with one fenestration closed and the other open (TRAAK-CO, PDB ID: 4I9W) and TWIK-1 with two fenestrations opened (TWIK-1-OO, PDB ID: 3UMK). Side chain modeling was performed with the Biased

Probability Monte Carlo (BPMC) method (*Abagyan and Totrov, 1994*). The models were embedded into a pre-equilibrated membrane in a periodic boundary condition box with water molecules and ions. Each system was subjected to minimization and 10 ns molecular dynamics (MD) simulation, as previously described (*Ramírez et al., 2017*). To resolve the fenestrations dimensions of the TASK-1 structures HOLE (*Smart et al., 1996*) was used in the last frame of the MD simulations. For each MD simulation, a total of 20 structures were saved (one snapshot each 0.5 ns) to perform the docking using the Glide v.7.4 module of the Schrödinger suite (*Friesner et al., 2004*).

## Classical and Induced Fit Docking experiments

All TASK-1 models were selected as docking targets. Ten conformers of bupivacaine were generated using ConfGen (*Watts et al., 2010*) to enhance the sampling of the ligand. One grid was generated in each fenestration, another grid in the central cavity and a fourth grid that includes the two fenestrations and the central cavity. Standard precision mode (SP) of Glide was used for docking and 8000 bupivacaine poses *per* TASK-1 model were obtained (100 poses *per* grid in each frame, 20 frames *per* model), except for TASK-1 based on TREK-2-CC, where only 2000 poses were obtained because fenestrations are closed. The Conformer Cluster script (available in www.schrodinger.com/scriptcenter/) was used to process and to organize the poses. The clustering was done by RMSD ($\leq 2$ Å) considering only the heavy atoms of the bupivacaine. Poses within significant clusters (*Bottegoni et al., 2006*) were chosen for calculation of the experimental interaction score (EIS), for which a score for each interaction between bupivacaine and the residues of the binding site for each pose was assigned, as previously described (*Ramírez et al., 2017*). For TPA and TEA molecular docking, a TASK-1 model based on TWIK-1 was aligned with the KcsA crystal structure (PDB ID: 1J95) (*Zhou et al., 2001*) and a docking procedure similar to that reported for channel TREK-1 was performed (*Piechotta et al., 2011*). Schrödinger's Induced Fit Docking (IFD) (*Sherman et al., 2006*) was performed using Glide v.7.4 (*Friesner et al., 2004*) with the standard precision mode. Here, for each pose, a Prime structure prediction adjusts the ligand by reorienting nearby side chains. First we generated 20 different starting structures (seeds) for each homology model, by taking a frame from an MD simulation every 0.5 ns. We performed induced fit dockings using the 20 different starting structures for each homology model. This reflects 40 or only 20 induced fit dockings per homology model if only one fenestration is in the open state. For the initial ligand sampling, docking grid was centered on a selection of the experimental hits residues of each fenestration. A van der Waals radii of 0.5 Å for protein and ligand was used and 20 poses were generated for each structure *per* opened fenestration. Residues within 5 Å of each docking solution were minimized using Prime module of Schrödinger suite (Prime, version 4.7, Schrödinger, LLC, New York, NY, 2017). Subsequently, using the poses that survived this first round of induced fit dockings, a redocking without vdW scaling was carried out and the best structure according to the induced fit docking score was generated.

## Molecular Dynamics simulations and RMSF calculation

TASK-1 models based on TWIK-1 alone and in complex with the bupivacaine best docking solution were subjected to a conjugate gradient energy minimization and MD simulations in Desmond v3.0 using OPLS-2005 (*Jorgensen et al., 2016*; *Kaminski et al., 2001*) force field. The TASK-1 channels, as well as the channel-ligand complex, were embedded into a POPC lipid bilayer and were solvated by an orthorhombic box of SPC water model, covering the whole surface of each system. Cl$^-$ were used as counter ions in order to neutralize the systems and 0.096 M KCl was added to match the concentration used in electrophysiological measurements. The temperature was maintained at 300 K, while pressure was kept at one atm, employing the Nose-Hoover thermostat method with a relaxation time of 1 ps using the MTK algorithm (*Martyna et al., 1994*). Each simulation was conducted with 2.5 fs time steps. Two MD simulations for each system were performed; the first 40 ns were executed applying a restraint spring constant of 0.5 kcal $\times$ mol$^{-1}$ $\times$ Å$^{-2}$ to the secondary structure of the receptor, then, the last frame was taken and a second non-restricted 100 ns–MD simulation was performed. Data for docking simulations and HOLE analysis were collected every 0.5 ns in the last 100 ns-MD simulation. To characterize local changes along TASK1, the Root Mean Square Fluctuation (RMSF) was calculated over the trajectory using the following equation:

$$RMSF_i = \sqrt{\frac{1}{T}\sum_{t=1}^{T}\left\langle r_i'(t) - r_i\left(t_{ref}\right)\right\rangle^2}$$

Where $T$ is the trajectory time over which the RMSF is calculated, $t$ is the reference time (the beginning of the MD simulation), $r$ is the position of residue $i$; $r^i$ is the position of atoms in residue $i$ after superposition on the reference, and the angle brackets indicate that the average of the square distance is taken over the selection of atoms in the residue.

## Ion solvation energy calculations

Ion solvation energy was calculated using the Adaptive Poisson-Boltzmann Solver (APBS) software (*Callenberg et al., 2010*) to calculate the Poisson-Boltzmann equation in an APBSmem interface (*Baker et al., 2001*). A TASK-1 model based on TWIK-1 was used for the continuum electrostatic calculations. The selectivity filter of the channel was aligned to the Z axis. The parameters used are summarized in *Supplementary file 1*.

## Quantification and statistical analysis

Every dataset was tested with a Shapiro-Wilk test for normality. Equality of variances was tested using either a parametric or non-parametric Levene's test. In case of similar variances, significance was probed using either a paired or unpaired Student's t-test and for not normally distributed data, a non-parametric Mann-Whitney U-test and a Wilcoxon signed-rank test for paired analyses was used, respectively. If the variances of the groups were different, significance was probed using either Welch's t-test and for not normally distributed data Mood's median test was used. Experiments were non-randomized and non-blinded and no pre-specified sample size was estimated. All data are presented as mean ± S.E.M. and all experiments were repeated from $N$ = 2–5 different batches (biological replicates). The number of experiments ($n$) as technical replicates are indicated in the subpanels of the Figures. Significances are indicated with *, p < 0.05; **, p < 0.01; ***, p < 0.001 in the Figures and the statistical method applied is provided in the respective Figure legend.

## Acknowledgements

We thank Oxana Nowak and Kirsten Ramlow for technical assistance. This work was supported by Deutsche Forschungsgemeinschaft (DFG) grant DE1482-4/1 to ND and Fondecyt Grant 1140624 to WG.

## Additional information

### Funding

| Funder | Grant reference number | Author |
|---|---|---|
| National Fund for Scientific and Technological Development | 1140624 | Wendy Gonzalez |
| Deutsche Forschungsgemeinschaft | DE1482-4/1 | Niels Decher |

The funders had no role in study design, data collection and interpretation, or the decision to submit the work for publication.

### Author contributions

Susanne Rinné, Data curation, Formal analysis, Investigation, Writing—original draft, Writing—review and editing; Aytug K Kiper, Data curation, Formal analysis, Validation, Investigation, Visualization, Writing—review and editing; Kirsty S Vowinkel, Data curation, Formal analysis, Investigation, Writing—review and editing; David Ramírez, Marcus Schewe, Formal analysis, Investigation, Visualization; Mauricio Bedoya, Phillip J Stansfeld, Software, Formal analysis, Investigation; Diana Aser, Michael F Netter, Formal analysis, Investigation; Isabella Gensler, Investigation; Thomas Baukrowitz, Formal

analysis, Supervision; Wendy Gonzalez, Software, Formal analysis, Supervision, Funding acquisition, Validation, Investigation, Visualization, Project administration; Niels Decher, Conceptualization, Resources, Data curation, Formal analysis, Supervision, Funding acquisition, Validation, Investigation, Visualization, Methodology, Writing—original draft, Project administration, Writing—review and editing

## Author ORCIDs
Aytug K Kiper (iD) http://orcid.org/0000-0003-2850-2523
Marcus Schewe (iD) http://orcid.org/0000-0002-6192-5651
Niels Decher (iD) http://orcid.org/0000-0001-8892-1231

## Decision letter and Author response
Decision letter https://doi.org/10.7554/eLife.39476.022
Author response https://doi.org/10.7554/eLife.39476.023

## Additional files

### Supplementary files
• Supplementary file 1. Parameters used to calculate the Poisson-Boltzmann equation for TASK-1.
DOI: https://doi.org/10.7554/eLife.39476.019
• Transparent reporting form
DOI: https://doi.org/10.7554/eLife.39476.020

### Data availability
All data generated or analysed during this study are included in the manuscript and supporting files.

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
