## [Decision Letter]

Thank you for sending your article entitled "The molecular basis for an allosteric inhibition of K^+^-flux gating in K_2P_ channels" for peer review at *eLife*. Your article is being evaluated by two peer reviewers, and the evaluation is being overseen by a Reviewing Editor and Richard Aldrich as the Senior Editor.

Given the list of essential revisions, including new experiments, the editors and reviewers invite you to respond within the next two weeks with an action plan and timetable for the completion of the additional work. We plan to share your responses with the reviewers and then issue a binding recommendation.

Reviewer #1:

The manuscript by Rinné et al. reports the discovery of the binding site of the local anesthetics bupivacaine to the two-pore-domain potassium channel TASK-1. A combination of electrophysiology and molecular modeling is used to characterize inhibition and to infer the structure of the drug-channel complex. The major finding of the paper is that bupivacaine binds to the "fenestrations", protein cavities whose volume decreases dramatically on transitioning from the "down" to the "up" state. The hypothesized selective affinity for the "down" state then rationalizes the inhibitory effect, since voltage-dependent opening of the channel preferentially occurs from the "up" state. The paper is well written and overall compelling. The conclusions are supported by a careful combination of experiments and calculations. Concerning the molecular modeling component, the calculations have been competently performed and rigorously analyzed. I recommend publication of this manuscript after the authors have addressed the following concerns.

1) the comparative homology modeling component is clearly crucial for the subsequent docking calculations. Especially in light of this relevance, the information about the modeling protocol provided in this manuscript is too scant. There are several aspects that need to be extensively described:

a) Choice of the template: several experimental structures have been preliminary explored, what is the phylogenetic relationship of all these genes with TASK-1? Are these all paralogs? Which one is the closest relative?

b) Side chain modeling: how was this done? This is a clearly relevant issue for subsequent docking and it is absolutely necessary for the reader to know what method was used and if more than one of the state of the art methods (e.g SCWRL) was compared.

2) Somewhat connected to the previous point is the docking protocol: given the degree of uncertainty about the side chains, it might be advisable to attempt some Induced Fit Docking, since in this approach side chains are reconstructed from scratch after the optimal docking pose has been found. This could eliminate some dependency on the initial model.

3) A crucial point supporting the allosteric mechanism is the observation from MD simulations that, without the drug, a channel starting in the "down" state transitions quickly top the "up" one (while this does not happen in presence of the drug). While I find the idea intriguing, there is little information in the manuscript about this conformational transition. A more descriptive and quantitative presentation of this result is necessary. In particular, the authors should describe the motion of the major structural elements of the channel in time to rationalize this "down" to "up" transition and compare the two cases, with and without the drug.

Reviewer #2:

This manuscript addresses the mechanism of action of a local anesthetic sodium channel blocker bupivacaine on TASK1/3 channels. The paper shows that the block differs in subtle but perhaps important ways from other recognized pore blockers, and the argument is made that the insights gained may lead to development of specific TASK blockers. Unfortunately, efforts to 'sell' the paper make it difficult to read, but also highlight logical weaknesses that seriously undermine the story.

1) In general, the writing style leads to some important misleads. In part, this may be a problem of literal translation from German to English (e.g. first paragraph of the Results section should be "Initially we aimed to investigate.… such as A21899.…", not as written. In the third sentence 'Speed' refers to distance travelled over time. You mean rate), but it also reflects an attempt to oversell the story, and to casually move from suggestion to acceptance. This is a problem throughout.

2) The authors stress an apparently different mechanism of action for bup vs the other blockers, but this seems a bit of a stretch. First, the voltage-dependence of block (Figure 1C) is extremely shallow – too shallow to calculate apparently, and there is a hint of voltage-dependence in the A1899 and A293 records, which could be more apparent at higher concentrations, or more negative voltages. Second, while pointing out novel residues identified as affecting block, the authors seem to ignore the relevance of key residues involved in A1899/A293 block (i.e. res 118, 235, 239 and 240) also being key to bup block. The authors also uncritically assume that these residues are directly involved in blocker binding, but the old saw remains true – action does not imply binding. It is conceivable that additional residues affect the block allosterically. Somewhere it would be beneficial to see the structures of each of the blockers. Finally, the authors highlight the apparent preferential inhibition of the 'activating' fraction of the current at positive voltages by bup, but not the other blockers. This fraction appears to be ~20% of the control current at +70mV, but this fraction seems not only to be lost for 500 μM bup, but also seems to be reduced by much lower concentrations of A1899 and A293.

3) In the process of explaining the results the authors make 2 novel suggestions: (1) that voltage dependent gating of TASK channels involves "down"-to-"up" conformational changes and closing of the fenestrations; and (2) that bupi binds within the fenestrations to exert its effect. The authors use alanine-scanning mutagenesis to probe the binding site for bup. Later, they propose that bup may prevent voltage-activation by binding within the fenestrations and preventing the down-to-up conformational changes involved in voltage-dependent activation of TASK channels (a new concept). In this model, bup binding must be state-dependent. So, any of the mutations identified as hits from the alanine scan could easily decrease drug sensitivity by shifting the conformational equilibrium or channel gating behaviour. In this regard it would be important to demonstrate that the most important/impactful mutations do not significantly alter intrinsic channel behavior. This idea is clearly illustrated in the data: Mutation of the pore helix residue T199 reduces inhibition. But if inhibition preferentially blocks voltage-dependent gating, and mutation of T199 abolishes voltage-dependent gating – then it follows that block will be affected, and interaction with this site is not proven.

In subsection “TASK-1 channels are fenestrated providing the bupivacaine binding site” they state "Here we noted that the best overlap with our functional data is given for docking solutions with bupivacaine located in the side fenestrations…". This approach seems biased. Rather than performing unbiased docking and then taking the most likely poses for testing, they have selected docking results to fit the previous data – how was "best overlap" determined? They also state that docking solutions increase as the size of the fenestration increases – this seems like a tautology. Nothing will bind if there is no room? Support for the fenestration binding site would be provided by showing the drug acts in a state-dependent manner on other K_2P_ channels, e.g. what effect does mechanoactivation of TRAAK/TREK channels have on the voltage-independent inhibition by bup?

In Results paragraph one the authors state that "varying the intracellular pH…did not alter the TASK-2 affinity". Use of "affinity" should be avoided for referred measures of effect, better to use "apparent affinity" or "sensitivity". More importantly, pHi does induce a significant shift in sensitivity. This is of course hidden somewhat by the log plot of IC_50_ in Figure 1R, but there is ~ 3 fold decrease in IC_50_ from neutral to pH 9, right around the predicted pKa8.1 of bup (https://pubchem.ncbi.nlm.nih.gov/compound/bupivacaine), although within the confines of the channel pore, free solution estimates may not be relevant anyway.

The authors present a model wherein voltage-dependent conformational changes move channels into the up state, and bup binding prevents this conformational change. If correct, then voltage-activation should decrease accessibility of the fenestration binding site for bup, and thereby decrease drug sensitivity. However, there is scant evidence to support the model that voltage-gating involves down-to-up transitions, and as the model for bup inhibition relies on this premise, the proposed paradigm breaks down: The authors draw analogies to TREK/TRAAK channels and suggest that bup may bind and prevent voltage-dependent gating by preventing down-to-up conformational changes (subsection “MD simulations – binding in the fenestrations prevents fenestration gating” and “Bupivacaine stabilizes the closed state of TASK channels”). However, for TREK-2, voltage-dependent activation, has no major effect on state-dependent inhibition by norfluoxetine – suggesting that voltage-dependent gating does not involve down-to-up transitions (McClenaghan et al., 2016). Furthermore, Schewe and colleagues previously showed that AA and stretch activation (which provoke down-to-up transitions: Aryal et al. 2017, McClenaghan et al., 2016, Brennecke et al., 2018) converts channels into non-voltage dependent, constitutively open states. Taken together, this suggests that up-to-down transition gating and voltage-dependent gating may be structurally distinct. Therefore it is not logically clear why bup binding and prevention of down-to-up transition would specifically affect voltage-dependent gating.

4) The above are concerns with the results and interpretations themselves. In the text describing them, there are many somewhat rhetorical, but potentially misleading statements that are apparently there to guide the reader to the authors' desired interpretation. For instance "These data argue against a contribution of the biophysical features, like the charge of the compound, to the voltage-dependent inhibition". First, the data being described don't fit – for TREK1, TRAAK and TASK2 the trend is also towards a qualitatively similar (if even weaker) voltage-dependence (Figure 1P). Second, it is not clear how the biophysical features, including the charge of the compound, could NOT contribute to the voltage-dependence? The authors state that "TASK1.… most likely reflecting increased currents due to the K flux gating at the selectivity filter.." and that ".. K_2P_ channels operate as a one-way check valve..". This is not the case – the current is for the most part linear around the reversal potential. And "..time-dependent current 'inactivation'" is not at all visible in Figure 1A. Then the authors conclude the first section with "Our data indicate that bupivacaine does not act as a classical open channel blocker", and "is utilizing a novel K_2P_ channel inhibiting mechanism impairing voltage-dependent K flux gating". At least the first point seems incorrect – everything points to a very weakly voltage-dependent block from somewhere below the selectivity filter – classical pore blocking. In this regard, 'trans knock-off' experiments assessing the ability of external [K] to influence the block would be informative.

5) In the Discussion there are many grammatical and rhetorical missteps, some of which really detract from the underlying potential, and should be carefully revised. e.g:

Sentence two should be "no crystallographic structures have been reported so far…" and "valuable predictions regarding the structure"

Sentence four:."Currently, the most likely model is.." is oxymoronic and a biased statement. 'Likely' means likely to be correct, but a model is never correct. Should read something like "A likely model is" or to be rally agnostic "One model is"

Discussion paragraph two: "TASK channels provide two side fenestrations as allosteric binding sites for the channel inhibition we describe here", and paragraph three "Bupivacaine located in the side fenestrations interacts with residues of the second pore helix. These sentences mix up so many interpretations, hopes, and potential falsehoods – all as fact, that I am not sure where to begin to parse them out.

Paragraph four: Finally "These drugs would have little or no effect on the membrane potential…" seems to indicate a lack of understanding. We are talking about K channels, and blocking them will affect the membrane potential. The voltage-dependence of the block itself is not the relevant issue.

[Editors' note: further revisions were requested prior to acceptance, as described below.]

Thank you for resubmitting your work entitled "The molecular basis for an allosteric inhibition of K^+^-flux gating in K_2P_ channels" for further consideration at *eLife*. Your revised article has been favorably evaluated by Richard Aldrich (Senior Editor), a Reviewing Editor, and two reviewers.

The authors have done a good job of addressing the major concerns of the reviewers and making it more readable. The manuscript has certainly improved a lot but there are some remaining issues that need to be addressed before acceptance, as outlined below:

1) The clarifications regarding the proposed model not claiming that the 'down' to 'up' transition is intrinsically voltage-dependent, that the authors conclude that bup prevents transition from 'down' to 'up' conformations, and that voltage-dependent gating preferentially occurs from the 'up' state, are appreciated. However, there is still some concern that the model is built upon a flawed premise. The authors are correct in saying that the Aryal et al., Brennecke and de Groot, and McClenaghan et al., papers suggest that the related TREK/TRAAK channels preferentially open from the up state (as stated in the Introduction and Discussion first paragraph), but these studies do not support the proposal that voltage-dependent activation preferentially occurs from the up-state. On the contrary, in TREK/TRAAK channels, mechanical activation involves down-to-up transitions, which Schewe et al., show involves conversion to the non-voltage-dependent "leak-mode". Therefore, the 'up' conformation will be less voltage-sensitive in TREK/TRAAK channels, which does not provide support for voltage-dependent activation of K_2P_ channels in general occurring preferentially from the up-state as proposed in the Introduction and Discussion paragraph one. Since some of the present authors are also key authors on the Cell 'leak-mode' paper, perhaps they can clarify this apparent contradiction.

2) Furthermore, the present model relies on TASK channels interconverting between down and up conformations, which in itself is still a novel idea and cannot be assumed to occur by simple analogy with related TREK/TRAAK channels. The MD simulations provide the only hint that such conformation changes occur, and there does not appear to be any experimental evidence. What do the authors propose controls this interconversion? Is this involved in normal channel gating/regulation? Interestingly, the authors state that a leak mode has never been reported for the TASK channels. Could this be because they never adopt an 'up' conformation?

3) In the absence of experimental evidence for 'down' to 'up' conformational changes (MD simulations aside), and the questionable logic that voltage-dependent gating occurs preferentially from the up state, the proposed model still falters. Accepting the extensive mutagenesis, docking, and new K^+^ knock-off experiments (a welcome addition), which do support the idea that bup binds within the fenestrations, is it not equally plausible (and applying Occam's razor, more so), that bup binding is more likely to allosterically prevent voltage-dependent selectivity filter gating (flux gating) without any need to invoke down-to-up conformational changes – as is actually suggested in Discussion paragraph four?

---

## [Author Response]

Reviewer #1:The manuscript by Rinné et al. reports the discovery of the binding site of the local anesthetics bupivacaine to the two-pore-domain potassium channel TASK-1. A combination of electrophysiology and molecular modeling is used to characterize inhibition and to infer the structure of the drug-channel complex. The major finding of the paper is that bupivacaine binds to the "fenestrations", protein cavities whose volume decreases dramatically on transitioning from the "down" to the "up" state. The hypothesized selective affinity for the "down" state then rationalizes the inhibitory effect, since voltage-dependent opening of the channel preferentially occurs from the "up" state. The paper is well written and overall compelling. The conclusions are supported by a careful combination of experiments and calculations. Concerning the molecular modeling component, the calculations have been competently performed and rigorously analyzed. I recommend publication of this manuscript after the authors have addressed the following concerns.

Thank you for reviewing the manuscript and for your positive and constructive comments. Please note that we (i) confirmed our initial docking results with the help of induced fit dockings, (ii) provided a descriptive and quantitative analysis of the transition from `down´ to `up´ state, and (iii) improved the methodical information regarding the template choosing, docking, and side chain modelling.

1) the comparative homology modeling component is clearly crucial for the subsequent docking calculations. Especially in light of this relevance, the information about the modeling protocol provided in this manuscript is too scant. There are several aspects that need to be extensively described:a) Choice of the template: several experimental structures have been preliminary explored, what is the phylogenetic relationship of all these genes with TASK-1? Are these all paralogs? Which one is the closest relative?

Thank you for bringing this to our attention. We have added the following sentence to the revised manuscript: “Genes encoding the crystallized TWIK and TREK/TRAAK channels are paralogs of KCNK3 encoding TASK‐1 and within these channels TWIK‐1 is the closest relative to TASK‐1 with a sequence identity of 26.1%. These genetic lines of evidence are further supported by the fact that TWIK‐1 and TASK‐1 share functional similarities, i.e. as they change their permeability upon extracellular acidification and as they form heterodimeric channels (Ma et al., 2012; Plant et al., 2012).”

b) Side chain modeling: how was this done? This is a clearly relevant issue for subsequent docking and it is absolutely necessary for the reader to know what method was used and if more than one of the state of the art methods (e.g SCWRL) was compared.

Thank you for identifying this additional lack of information. We did not test more than one of the state of the art method in our initial approach without induced fit dockings and included the necessary information in the revised Materials and methods section. Here we now state: “Side chain modeling was performed with the Biased Probability Monte Carlo (BPMC) method (Abagyan and Totrov, 1994, PMID: 8289329).”

2) Somewhat connected to the previous point is the docking protocol: given the degree of uncertainty about the side chains, it might be advisable to attempt some Induced Fit Docking, since in this approach side chains are reconstructed from scratch after the optimal docking pose has been found. This could eliminate some dependency on the initial model.

Thank you for this excellent advice. We performed Schrödinger's Induced Fit Docking (IFD) by docking the active ligand with Glide (see revised Materials and methods section). Indeed the suggested approach reduced some dependency from the initial model, leading to docking solutions in the side fenestrations of all homology models that are highly similar to that we reported in the initial manuscript. Thus, our IFD experiments confirmed the binding site we initially proposed.

In our IFD approach we first generated 20 different starting structures (seeds) for each homology model, by taking a frame from a MD simulation every 0.5 ns. We performed IFDs using the 20 different starting structures for each homology model. This reflects 40 or only 20 IFDs per homology model if only one fenestration is in the open state. IFD provides, only if a successful docking is performed, the best binding pose. In the TASK‐1 model based on TWIK‐1‐OO, IFD successfully found a binding pose in all the 40 fenestrations of the 20 structures, showing the highest success rate of the IFD dockings (Figure 4—figure supplement 2A). As the highest success rate was given in the TWIK‐1‐based homology model and TWIK‐1 is the closest relative of the crystallized K_2P_channels, we selected the pose with the lowest IFD energy that is also consistent with the TEA versus TPA competition experiments, an approach that we have also chosen in the manuscript for the classical docking experiments. This IFD solution in the TWIK‐1‐OO based TASK‐1 homology model is very similar to the one obtained by classical docking (Figure 4—figure supplement 2B‐C).

In another set of experiments that was not included in the revised manuscript, we grouped all the poses obtained in the four models using the conformer cluster script (www.schrodinger.com/scriptcenter/) following a structural alignment between the homology models. The clustering was performed by RMSD (≤2Å) considering only the heavy atoms of bupivacaine. Within the significant clusters (Bottegoni et al., 2006), we selected the pose with the lowest IFD *per* cluster, which was consistent with the TEA and TPA experiments, as described above. Using this approach, we identified only one cluster where the pose with the lowest IFD was consistent with the TEA and TPA experiments. Strikingly this pose is the same docking solution as presented in the new Figure 4—figure supplement 2. The different clusters obtained in the side fenestrations of the different homology models support however your initial expectation that the IFD approach reduced some dependency from the initial model (not illustrated).

3) A crucial point supporting the allosteric mechanism is the observation from MD simulations that, without the drug, a channel starting in the "down" state transitions quickly top the "up" one (while this does not happen in presence of the drug). While I find the idea intriguing, there is little information in the manuscript about this conformational transition. A more descriptive and quantitative presentation of this result is necessary. In particular, the authors should describe the motion of the major structural elements of the channel in time to rationalize this "down" to "up" transition and compare the two cases, with and without the drug.

Thank you for this very constructive comment. Accordingly, to have a more descriptive and quantitative presentation of the `down´ to `up´ transition, we did further analysis of the MDs with and without bupivacaine and displayed these findings in the revised Results section, Figure 4G‐J and the novel Video 1. We now state in the revised Results section:

“This transition of the channel from the `down´ to `up´ state which exclusively occurs in the absence of bupivacaine, is caused by a coordinated movement of the M2, M3 and M4 segments (Figure 4G,H and Video 1), similar as previously reported for 100 ns MDs of TREK‐2 channels (Dong et al., 2015). These movements in the absence of bupivacaine are also reflected by an analysis of the root‐mean‐square fluctuations (RMSF) (Figure 4I), revealing that the M2‐M3 linker and the M4 segment are much more flexible during MD simulations in the absence of bupivacaine. As these movements are expected to concomitantly close the side fenestrations, we have quantitatively analyzed this transition, monitoring the bottleneck diameter of the fenestration in MD simulations over time (Figure 4j). Here the bottleneck radius of the fenestrations quickly and significantly decreases in the absence of bupivacaine, while bupivacaine stabilizes the bottleneck radius of the side fenestrations to remain at about 3 Å over the whole time period of the 100 ns MD simulations (Figure 4J).”

Reviewer #2:This manuscript addresses the mechanism of action of a local anesthetic sodium channel blocker bupivacaine on TASK1/3 channels. The paper shows that the block differs in subtle but perhaps important ways from other recognized pore blockers, and the argument is made that the insights gained may lead to development of specific TASK blockers. Unfortunately, efforts to 'sell' the paper make it difficult to read, but also highlight logical weaknesses that seriously undermine the story.

Thank you for reviewing our manuscript. Obviously there were misunderstandings in the way you understood our manuscript, i.e. we do not claim that the `down´ to `up´ transition is voltage‐dependent. We hope that we have convinced you with the explanations of our data provided in the point‐by‐point response in such a way that the manuscript is now more comprehensive and the interpretations are easier to follow, so you do not get the impression anymore that we want to `sell´ something non‐conclusive.

1) In general, the writing style leads to some important misleads. In part, this may be a problem of literal translation from German to English (e.g. first paragraph of the Results section should be "Initially we aimed to investigate.… such as A21899.…", not as written. In the third sentence 'Speed' refers to distance travelled over time. You mean rate), but it also reflects an attempt to oversell the story, and to casually move from suggestion to acceptance. This is a problem throughout.

First of all, we would like to apologize for the unprecise expressions which we have carefully removed from the manuscript as you suggested. However, please note that the other Referee does not share your opinion that we attempted to `oversell´ the story, for instance reviewer #1 stated: “The paper is well written and overall compelling.” We agree that the writing style needed improvements and we are thankful for your suggestions how to achieve this. Thus, we corrected all the problematic phrases you identified.

2) The authors stress an apparently different mechanism of action for bup vs the other blockers, but this seems a bit of a stretch. First, the voltage-dependence of block (Figure 1C) is extremely shallow – too shallow to calculate apparently, and there is a hint of voltage-dependence in the A1899 and A293 records, which could be more apparent at higher concentrations, or more negative voltages.

We agree that the voltage‐dependence is shallow, however it is similar to classical voltage‐dependent blockers, i.e. as observed for voltage‐dependent Kv1.5 blockers (i.e. Longobardo et al., 2000, PMID: 10807678). We do not agree that the voltage‐dependence is to shallow to calculate and the quantification that we have provided clearly showed a highly significant (p<0.001) voltage‐dependent block by bupivacaine, with a 2.5‐fold more pronounced inhibition at +70 mV compared to ‐70 mV, as analyzed in Figure 1D. In contrast, A1899 and A293 did not show a hint of voltage‐dependence or even significant differences in inhibition over this wide voltage range (Figure 1I,N). In the revised manuscript, we have also included the respective statistics in the panels of Figure 1C,H,M. In addition, the concentrations were well chosen and comparable for all three drugs, as these were consistently chosen to approximately cause sixty percent inhibition at potentials around 0 mV, to optimally resolve putative differences occurring with more and/or less pronounced inhibition. Thus, we think that increasing the concentrations will not help to further resolve a voltage‐dependence. As we did not observe significant changes in the quantification probing voltage‐dependent inhibition by A1899 and A293 (Figure 1I,N), we think that there is no need to further extent the voltage range of our experiments which also seems sufficient with a Δ of 140 mV.

A Woodhull analysis for bupivacaine inhibition of TASK‐1 yields values similar to that of weak voltage‐dependent blockers as noted above, while we could not successfully fit the data of A1899 and A293. However, we did not provide a δ‐value for bupivacaine in TASK‐1 in the manuscript, as the Woodhull‐model assumes that only one charged molecule binds in the central pore of a channel (Woodhull, 1973, PMID: 4541078). Thus, this model is not sufficient to explain our complex binding mode in which the drug is not positioned in the central cavity, as two molecules might sense the voltage and as our data clearly indicate that it is not exclusively the charged molecule that causes the inhibition. Please also note that this analysis does not provide a quantification of the extent of voltage‐dependence, but rather makes predictions towards the fraction of the transmembrane potential field through which the blockers move to reach their binding site (Woodhull, 1973, PMID: 4541078). As we have already mapped the binding site and as this is located outside the water filled central cavity, we do not think that providing data of a Woodhull analysis would improve the manuscript.

Second, while pointing out novel residues identified as affecting block, the authors seem to ignore the relevance of key residues involved in A1899/A293 block (i.e. res 118, 235, 239 and 240) also being key to bup block. The authors also uncritically assume that these residues are directly involved in blocker binding, but the old saw remains true – action does not imply binding. It is conceivable that additional residues affect the block allosterically. Somewhere it would be beneficial to see the structures of each of the blockers.

Please note that your statements about the key residues involved in A1899/A293 block (i.e. res 118, 235, 239 and 240) are not fully correct, at least for A293. Firstly, the A293 binding site was not yet reported/published and secondly, it is different to that of A1899 which we can clearly state, as we have mapped the residues relevant for A293 inhibition (manuscript currently under revision). A293 does not involve all key residues of A1899, i.e. mutating the key residue I235 leaves A293 inhibition unaffected. Nevertheless, we have included a small section explaining the partial overlap in the A1899 and bupivacaine binding site, as it is important to explain to the reader that many of the A1899 binding residues are not strictly `pore facing´ (Streit et al., 2011) and are likely to be also accessible from the side fenestrations (Ramirez et al.,2017). This section, together with the fact that we were able to verify contacts of bupivacaine with these residues in MDs, should prevent misinterpretations that we simply ignored `pore facing´ residues for bupivacaine binding.

Thank you for the excellent point that we should more carefully discriminate between binding and action which we carefully checked throughout the manuscript. In this context, we have also included a small Discussion section on allosteric effects.

The new Discussion section dealing with the two points raised above reads:

“While we identified specific residues to exclusively affect block by bupivacaine and not A1899, there is still a partial overlap to the binding site of the open channel blocker A1899 (i.e. residues I118, I235, G236, L239 and N240). This apparent discrepancy can be explained by the fact that many of the A1899 binding residues (Streit et al., 2011) are not strictly `pore facing´ (Streit et al., 2011) and are likely to be also accessible from the side fenestrations (Ramirez et al., 2017). In general, one has to consider that reduced inhibition of a mutant does not necessarily imply binding at the respective site, as it is conceivable that the mutation affects the block allosterically. This caveat has to be specifically considered when studying mutations that might simultaneously interfere with drug binding and the gating mechanism, like in the case of T199C (Schewe et al., 2016). However, the fact that we conclusively mapped a binding site that includes a large set of residues and that this site was confirmed by in silico dockings, MD simulations and TEA/TPA competition experiments, must lead to the conclusion that we have identified the most likely binding mode for bupivacaine.”

Thank you for the suggestion to illustrate the chemical structures of bupivacaine, A1899 and A293 which are now included in the revised Figure 1 (Figure 1A, F and K).

Finally, the authors highlight the apparent preferential inhibition of the 'activating' fraction of the current at positive voltages by bup, but not the other blockers. This fraction appears to be ~20% of the control current at +70mV, but this fraction seems not only to be lost for 500 μM bup, but also seems to be reduced by much lower concentrations of A1899 and A293.

Thank for discussing this point. As you have got the wrong impression that also A1899 and A293 seem to affect the activating fraction of the current, we have quantified the inhibition of the K^+^‐flux gated component which is now displayed in the new Figure 1—figure supplement 3. Here we can now more clearly show, that the `activating´ fraction is only significantly reduced for bupivacaine, but not for A1899 and A293. In these experiments lower concentrations of A1899 and A293 were used in order that all compounds cause a similar extent of inhibition at potentials around 0 mV, as discussed above.

3) In the process of explaining the results the authors make 2 novel suggestions: (1) that voltage dependent gating of TASK channels involves "down"-to-"up" conformational changes and closing of the fenestrations; and (2) that bupi binds within the fenestrations to exert its effect. The authors use alanine-scanning mutagenesis to probe the binding site for bup. Later, they propose that bup may prevent voltage-activation by binding within the fenestrations and preventing the down-to-up conformational changes involved in voltage-dependent activation of TASK channels (a new concept). In this model, bup binding must be state-dependent. So, any of the mutations identified as hits from the alanine scan could easily decrease drug sensitivity by shifting the conformational equilibrium or channel gating behaviour. In this regard it would be important to demonstrate that the most important/impactful mutations do not significantly alter intrinsic channel behavior. This idea is clearly illustrated in the data: Mutation of the pore helix residue T199 reduces inhibition. But if inhibition preferentially blocks voltage-dependent gating, and mutation of T199 abolishes voltage-dependent gating – then it follows that block will be affected, and interaction with this site is not proven.

Thank you for this careful discussion of our data. However there is a misinterpretation of our statements, as we did not propose a new concept of channel gating with a voltage‐dependent transition from the `down´ to the `up´ state. Although we propose that bupivacaine binds within the fenestrations to exert its effect, we do not state that the `down´ to `up´ state conformational change is voltage‐dependent. On the contrary, we proposed that binding of bupivacaine in the fenestration sterically prevents the `down´ to `up´ transition. However, as the voltage‐dependent opening of the selectivity filter (Schewe et al., 2016) preferentially occurs from the `up´ state (Brennecke and de Groot, 2018; McClenaghan et al., 2016), we observe a preferential inhibition of the voltage‐activated outward currents, as the precondition of the channel to be in the `up´ state is reduced (see Figure 6C‐D). As our theory was not sufficiently explained in the Results section, we provided a more detailed interpretation of our data in the revised Discussion section (paragraph one).

We agree that it is crucial to check whether mutants introduced by site directed mutagenesis exhibit altered kinetical properties, to avoid false positive hits in alanine scanning approaches. For this purpose we showed in the initial manuscript currents of the A114V, F194A und L239A mutations. Nevertheless, we found that the residues we identified in our alanine scan do not show strikingly different kinetical features. Also the T93C and T199C mutants described by Schewe et al. do not show unusual current or channel behavior in our hands using TEVC recordings. For sure alanine mutations in our scan have variable kinetics, however for the crucial hits in the pore helix with T93C, T199C and F194A we observed changes in the current kinetics which are within the normal variance.

Nevertheless, we agree to your point that point mutations can cause indirect effects and thus we included a small Discussion section about the allosteric caveats when studying mutations that might simultaneously interfere with drug binding and gating mechanism. This new Discussion section has been already cited in our response to your previous comments (2) about allosteric effects. The relevant part of this new Discussion section reads:

“In general, one has to consider that reduced inhibition of a mutant does not necessarily imply binding at the respective site, as it is conceivable that the mutations affects the block allosterically. This caveat has to be specifically considered when studying mutations that might simultaneously interfere with drug binding and the gating mechanism, like in the case of T199C (Schewe et al., 2016). However, the fact that we conclusively mapped a binding site that includes a large set of residues and that this site was confirmed by in silico dockings, MD simulations and TEA/TPA competition experiments, must lead to the conclusion that we have identified the most likely binding mode for bupivacaine.”

In subsection “TASK-1 channels are fenestrated providing the bupivacaine binding site” they state "Here we noted that the best overlap with our functional data is given for docking solutions with bupivacaine located in the side fenestrations…". This approach seems biased. Rather than performing unbiased docking and then taking the most likely poses for testing, they have selected docking results to fit the previous data – how was "best overlap" determined?

Selection of the best docking solution was performed in an unbiased manner, as explained in the initial manuscript. Briefly, docking poses within significant clusters were analysed using MM‐GBSA and the most stable docking solutions were generated using the TWIK‐based TASK‐1 homology model. Within these poses, the experimental interaction score (EIS), a score for each interaction between bupivacaine and the residues of the binding site for each pose, was assigned (Ramirez et al., 2017). The best overlap with the functional data (the best EIS) is given for a docking solution of bupivacaine located in the side fenestrations of TWIK‐1.

As reviewer 1 suggested, we performed Induced Fit Docking (IFD) experiments. Therefore, we performed Schrödinger's IFD by docking the active ligand with Glide (see revised Materials and methods section). Indeed the suggested approach confirmed the binding site we initially proposed.

In our IFD approach we first generated 20 different starting structures (seeds) for each homology model, by taking a frame from a MD simulation every 0.5 ns. We performed IFDs using the 20 different starting structures for each homology model. This reflects 40 or only 20 IFDs per homology model if only one fenestration is in the open state. IFD provides, only if a successful docking is performed, the best binding pose. In a TASK‐1 model based on TWIK‐1‐OO, IFD successfully found a binding pose in all the 40 fenestrations of the 20 structures, showing the highest success rate of the IFD dockings (Figure 4—figure supplement 2A). As the highest success rate was given in the TWIK‐1‐based homology model and TWIK‐1 is the closest relative of the crystallized K_2P_channels, we selected the pose with the lowest IFD energy that is also consistent with the TEA versus TPA competition experiments, an approach that we have also chosen in the manuscript for the classical docking experiments. This, IFD solution in the TWIK‐1‐OO based TASK‐1 homology model is very similar to the one obtained by classical docking (Figure 4—figure supplement 2B‐C).

They also state that docking solutions increase as the size of the fenestration increases – this seems like a tautology. Nothing will bind if there is no room? Support for the fenestration binding site would be provided by showing the drug acts in a state-dependent manner on other K_2P_ channels, e.g. what effect does mechanoactivation of TRAAK/TREK channels have on the voltage-independent inhibition by bup?

We agree that nothing can bind if there is no room. However, the fact that there is enough space does not conversely mean that a compound is likely to bind there. For instance A1899 and A293 do not bind in the side fenestrations even when they are in the open state. Strikingly, bupivacaine, unlike A1899 and A293, has a preference to bind to theses fenestrations once they are in the open state.

We do not agree that showing a state‐dependent block in other K_2P_channels supports our observation of a binding of bupivacaine to the side fenestrations of TASK‐1. Please note that the fact that bupivacaine does not cause a voltage‐dependent inhibition of all K_2P_channels is more in line with our hypothesis that it is not the charged bupivacaine in the central cavity which is causing the voltage‐dependent inhibition of TASK‐1. Conversely, we have identified a binding site unique to the TASK‐1/3/5 family and consistently we only get a significant voltage‐dependent block within this family.

As we did not observe a voltage‐dependent inhibition of channels in the TREK/TRAAK family we did not perform experiments on the influence of bupivacaine on the mechano‐activation of TRAAK/TREK channels.

In Results paragraph one the authors state that "varying the intracellular pH…did not alter the TASK-2 affinity". Use of "affinity" should be avoided for referred measures of effect, better to use "apparent affinity" or "sensitivity". More importantly, pHi does induce a significant shift in sensitivity. This is of course hidden somewhat by the log plot of IC_50_ in Figure 1R, but there is ~ 3 fold decrease in IC_50_ from neutral to pH 9, right around the predicted pKa8.1 of bup (https://pubchem.ncbi.nlm.nih.gov/compound/bupivacaine), although within the confines of the channel pore, free solution estimates may not be relevant anyway.

Thank you for the suggestions how to improve the manuscript. We followed your suggestion and replaced `affinity´ by `sensitivity´ where appropriate.

Please note that we did not intend to hide effects, when we considered that using a logarithmic scale would be a suitable way to display effects by changes in pH values. In fact you were able to note the approximately 3‐fold decrease in IC_50_. Considering the pKs of bupivacaine in free solution, a change from pH 5 to 9 would however mean a change of 99.9% charged bupivacaine to less than 10% of the charged compound. Albeit we agree that it is difficult to ultimately predict the pKs of bupivacaine within a protein environment, the observed three‐fold change in IC_50_ over this large pH range appears far too little to claim that it is primarily the charged compound mediating the channel inhibition and thus the voltage‐dependence of block.

As also discussed in the first version of the manuscript – if it is really primarily the charged bupivacaine causing a classical voltage‐dependent pore block, this voltage‐dependence depending on the physicochemical features of bupivacaine should, at a given pH, occur also in the other bupivacaine‐sensitive K_2P_channels.

The authors present a model wherein voltage-dependent conformational changes move channels into the up state, and bup binding prevents this conformational change. If correct, then voltage-activation should decrease accessibility of the fenestration binding site for bup, and thereby decrease drug sensitivity. However, there is scant evidence to support the model that voltage-gating involves down-to-up transitions, and as the model for bup inhibition relies on this premise, the proposed paradigm breaks down:

This is the same misunderstanding about the mechanism as discussed in one of the previous points, as we do not claim that the `down´ to `up´ state conformational change is voltage‐dependent. As our theory was not sufficiently explained in the Results section, we provided a more detailed interpretation of our data in the revised Discussion section, reading:

“We propose that binding of bupivacaine in the fenestration sterically prevents the `down´ to `up´ transition. However, as the voltage‐dependent opening of the selectivity filter (Schewe et al., 2016) preferentially occurs from the `up´ state (Brennecke and de Groot, 2018; McClenaghan et al., 2016), we observe a preferential inhibition of the voltage‐activated outward currents, as the precondition of the channel to be in the `up´ state is reduced (see Figure 6C‐D).”

The authors draw analogies to TREK/TRAAK channels and suggest that bup may bind and prevent voltage-dependent gating by preventing down-to-up conformational changes (subsection “MD simulations – binding in the fenestrations prevents fenestration gating” and “Bupivacaine stabilizes the closed state of TASK channels”). However, for TREK-2, voltage-dependent activation, has no major effect on state-dependent inhibition by norfluoxetine – suggesting that voltage-dependent gating does not involve down-to-up transitions (McClenaghan et al., 2016).

Although we did not claim that the `down´ to `up´ transition is voltage‐dependent in TASK‐1 (see points above), we agree that it is on a first thought puzzling that drug binding to somewhat similar binding sites can result in a voltage‐dependent block in one class of ion channels but not in another. Although it appears that the binding site of norfluoxetine in TREK‐1 is quite similar to that of bupivacaine in TASK‐1, it is possible that norfluoxetine does not as efficiently prevent the `down´ to `up´ transition in TREK‐1. In addition, it is known for ion channel blockers that subtle changes in the binding mode of different substances can already lead to strong changes in the kinetics of block, i.e. in the case of Kv1.5 blockers (Strutz‐Seebohm et al., 2007, PMID: 17982261; Decher et al., 2004,; Decher et al., 2006, PMID: 16835355).

Furthermore, Schewe and colleagues previously showed that AA and stretch activation (which provoke down-to-up transitions: Aryal et al. 2017, McClenaghan et al., 2016, Brennecke et al., 2018) converts channels into non-voltage dependent, constitutively open states. Taken together, this suggests that up-to-down transition gating and voltage-dependent gating may be structurally distinct. Therefore it is not logically clear why bup binding and prevention of down-to-up transition would specifically affect voltage-dependent gating.

As discussed above and re‐explained in more detail in the revised Discussion section, we did not claim that bupivacaine acts by preventing a voltage‐dependent `down´ to `up´ transition. In contrast, our model is in perfect agreement with the studies by McClenaghan et al., 2016 and Brennecke and de Groot, 2018 in terms that up‐todown transition gating and voltage‐dependent gating are most likely structurally distinct. Please note that the non‐voltage dependent leak modus proposed by Schewe et al. 2016, was not further supported by the subsequent studies of Aryal et al., 2017, McClenaghan et al., 2016, or Brennecke and de Groot, 2018. In addition, a leak mode has never been reported for TASK channels. As mentioned above the mechanism of action has been phrased more clearly in the revised Discussion section:

“We propose that binding of bupivacaine in the fenestrations sterically prevents the `down´ to `up´ transition. However, as the voltage‐dependent opening of the selectivity filter (Schewe et al., 2016) preferentially occurs from the `up´ state (Brennecke and de Groot, 2018; McClenaghan et al., 2016), we observe a preferential inhibition of the voltage‐activated outward currents, as the precondition of the channel to be in the `up´ state is reduced (see Figure 6C‐D).”

4) The above are concerns with the results and interpretations themselves. In the text describing them, there are many somewhat rhetorical, but potentially misleading statements that are apparently there to guide the reader to the authors' desired interpretation. For instance "These data argue against a contribution of the biophysical features, like the charge of the compound, to the voltage-dependent inhibition". First, the data being described don't fit – for TREK1, TRAAK and TASK2 the trend is also towards a qualitatively similar (if even weaker) voltage-dependence (Figure 1P).

We are somewhat disturbed by the referees opinion that we should not guide the reader to understand our interpretations while on the contrary he himself considers somewhat rhetorical or misleading statements when he is referring to or interpreting non‐significant baseline variations. We strongly object to this point, as there are only minor, no or only minimal trends, but not a significant voltage‐dependent block of other K_2P_channels (Figure 1P). Thus, the referee´s statement is not correct! The voltage‐dependent inhibition cannot reflect an unspecific open channel block of all K_2P_channels which is mediated exclusively by charged/protonated molecules, while one type of K_2P_channel is blocked in a voltage‐dependent manner with a p<0.001 and the other channels at the same pH show only non‐significant or not voltage‐dependent changes at all. Yet, we agree that the sentence was not phrased clearly enough and we have improved it to avoid misunderstandings. It now reads: “These data argue against a sole contribution of the protonation state and that only charged molecules cause the voltage-dependent inhibition”.

Second, it is not clear how the biophysical features, including the charge of the compound, could NOT contribute to the voltage-dependence? The authors state that "TASK1.… most likely reflecting increased currents due to the K flux gating at the selectivity filter.." and that ".. K_2P_ channels operate as a one-way check valve..". This is not the case – the current is for the most part linear around the reversal potential. And "..time-dependent current 'inactivation'" is not at all visible in Figure 1A.

The biophysical features do of course influence affinity and kinetics of most blockers. However, if the blocker is not sensing the electrical field or is not able to move because of the electrical field, then it is not exclusively the charged compound that blocks the channel. This idea is supported by our unusual binding site and that the IC_50_ of the drug was not strongly depending on the pHi and the protonation state of the blocker.

As discussed above we have replaced the sentence about the biophysical features. It now reads: “These data argue against a sole contribution of the protonation state and that only charged molecules cause the voltage-dependent inhibition”.

We were basically referring to the study by Schewe et al. and to our recordings in which we observed a decay of the inward currents over time when stepping to hyperpolarized potentials. However, we agree to your point that the current voltage relationship is for the most part linear around the reversal potential and thus deleted the statement about the one way check valve from the manuscript.

Then the authors conclude the first section with "Our data indicate that bupivacaine does not act as a classical open channel blocker", and "is utilizing a novel K_2P_ channel inhibiting mechanism impairing voltage-dependent K flux gating". At least the first point seems incorrect – everything points to a very weakly voltage-dependent block from somewhere below the selectivity filter – classical pore blocking. In this regard, 'trans knock-off' experiments assessing the ability of external [K] to influence the block would be informative.

We strongly disagree to your statement that everything points towards a classical pore block. Both, the mapping of the drug binding site with alanine mutagenesis screens and the molecular docking experiments show that the binding site is different to that of classical TASK open channels blockers. This novel binding site and the novel mechanism of inhibition are supported by the TEA versus TPA competition experiments, the differences in the ion solvation free energy profiles comparing charged TEA with charged or neutral bupivacaine located at its binding site and by the single channel analyses that clearly show difference in block mechanism between TEA and bupivacaine. With all these data pointing to a non‐classical pore plugging mechanism the most straightforward interpretation is an allosteric block mechanism. This allosteric block mechanism was subsequently successfully probed with molecular modelling experiments and put into a hypothesis/model that explains our experimental findings of primarily reduced outward currents, while at the same time implementing the current knowledge about K_2P_channel gating.

As requested, we performed `trans knock‐off´ experiments which strongly support the accumulated data that conclusively indicate that bupivacaine is not a classical open channel blocker of TASK‐1 channels (Results paragraph two and new Figure 1—figure supplement 4). We have previously reported that TASK‐1 inhibition by the open channel blocker A1899 is attenuated by extracellular potassium (Streit et al., 2011). Strikingly, while increasing extracellular potassium concentration gradually decreased TASK‐1 block by A1899, the inhibition of TASK‐1 by bupivacaine was not affected (Figure 1—figure supplement 4).

5) In the Discussion there are many grammatical and rhetorical missteps, some of which really detract from the underlying potential, and should be carefully revised. e.g:Sentence two should be "no crystallographic structures have been reported so far…" and "valuable predictions regarding the structure"Sentence four: "Currently, the most likely model is.." is oxymoronic and a biased statement. 'Likely' means likely to be correct, but a model is never correct. Should read something like "A likely model is" or to be rally agnostic "One model is"

Thank you. We have fixed the three sentences.

Discussion paragraph two: "TASK channels provide two side fenestrations as allosteric binding sites for the channel inhibition we describe here", and paragraph three "Bupivacaine located in the side fenestrations interacts with residues of the second pore helix. These sentences mix up so many interpretations, hopes, and potential falsehoods – all as fact, that I am not sure where to begin to parse them out.

We hope that after we provided in our revised manuscript the additional data and with our response the necessary explanation that you agree that these sentences where maybe phrased to strong but yet represent the most likely interpretations of the data. Thus, we have changed the first sentence it now reads: “two side fenestrations as putative allosteric binding sites for….” and the “most likely”

Paragraph four: Finally "These drugs would have little or no effect on the membrane potential…" seems to indicate a lack of understanding. We are talking about K channels, and blocking them will affect the membrane potential. The voltage-dependence of the block itself is not the relevant issue.

Thank you. However, we agree that this idea was not phrased well enough. The theory discussed in the manuscript was: as the block is voltage‐dependent and primarily occurring at depolarized potentials, a novel class of voltage‐dependent blockers that utilize this mechanism and only block at depolarized potentials, should leave the potassium channels unbiased and thus the membrane potential, as long as the membrane potential constantly remains hyperpolarized. As we do not intent to confuse the reader, obviously as we did here, and as we did not provide experimental evidence for this theory, we have removed the paragraph from the Discussion section.

[Editors' note: further revisions were requested prior to acceptance, as described below.]

The authors have done a good job of addressing the major concerns of the reviewers and making it more readable. The manuscript has certainly improved a lot but there are some remaining issues that need to be addressed before acceptance, as outlined below:1) The clarifications regarding the proposed model not claiming that the 'down' to 'up' transition is intrinsically voltage-dependent, that the authors conclude that bup prevents transition from 'down' to 'up' conformations, and that voltage-dependent gating preferentially occurs from the 'up' state, are appreciated. However, there is still some concern that the model is built upon a flawed premise. The authors are correct in saying that the Aryal et al., Brennecke and de Groot, and McClenaghan et al., papers suggest that the related TREK/TRAAK channels preferentially open from the up state (as stated in the Introduction and Discussion first paragraph), but these studies do not support the proposal that voltage-dependent activation preferentially occurs from the up-state. On the contrary, in TREK/TRAAK channels, mechanical activation involves down-to-up transitions, which Schewe et al., show involves conversion to the non-voltage-dependent "leak-mode". Therefore, the 'up' conformation will be less voltage-sensitive in TREK/TRAAK channels, which does not provide support for voltage-dependent activation of K_2P_ channels in general occurring preferentially from the up-state as proposed in the Introduction and Discussion paragraph one. Since some of the present authors are also key authors on the Cell 'leak-mode' paper, perhaps they can clarify this apparent contradiction.

Thank you for this discussion and criticism. We agree that we cannot finally prove the novel model that we have initially proposed and that we should not concomitantly transfer this novel model to a channel of another family. After I have discussed all the data again with Thomas Baukrowitz, we agree that our initial model might be too farfetched and we have chosen a more moderate model. We have changed the revised manuscript accordingly and do not claim anymore that these channels must have gated fenestrations and that preventing the gating of the fenestration must be the principle mechanism of action. We also do not claim anymore that TASK channels must exist in `up´ and `down´ states and that the `up´ state is the state from which voltage‐dependent opening preferentially occurs.

We followed your suggestion of applying Occam's razor and only state that we identified a novel allosteric mechanism of TASK inhibition resulting from binding of bupivacaine to lateral fenestrations of TASK‐1 and that bupivacaine reduces the voltage‐dependent K^+^‐flux gating leading to a weak voltage-dependent inhibition. The Figure 6 illustrating the mechanism of action was corrected accordingly.

2) Furthermore, the present model relies on TASK channels interconverting between down and up conformations, which in itself is still a novel idea and cannot be assumed to occur by simple analogy with related TREK/TRAAK channels. The MD simulations provide the only hint that such conformation changes occur, and there does not appear to be any experimental evidence. What do the authors propose controls this interconversion? Is this involved in normal channel gating/regulation? Interestingly, the authors state that a leak mode has never been reported for the TASK channels. Could this be because they never adopt an 'up' conformation?

We changed the model in the revised manuscript and do not claim anymore that TASK channels must have gated fenestrations and `up´ and `down´ states. The section about the MD simulations now finishes as followed:

“However, whether TASK‐1 channels gate in a similar way as TREK‐1 channels and whether they physiologically convert into an `up´‐state like conformation currently remains unknown. Thus, it also remains an open question whether effects of bupivacaine on the architecture of the fenestrations contribute to the allosteric inhibition mechanism.”

3) In the absence of experimental evidence for 'down' to 'up' conformational changes (MD simulations aside), and the questionable logic that voltage-dependent gating occurs preferentially from the up state, the proposed model still falters. Accepting the extensive mutagenesis, docking, and new K^+^ knock-off experiments (a welcome addition), which do support the idea that bup binds within the fenestrations, is it not equally plausible (and applying Occam's razor, more so), that bup binding is more likely to allosterically prevent voltage-dependent selectivity filter gating (flux gating) without any need to invoke down-to-up conformational changes – as is actually suggested in Discussion paragraph four?

As pointed out in point 1 and 2 we now follow your suggestion and propose that bupivacaine binding to the fenestrations is likely to allosterically prevent the voltage‐dependent selectivity filter gating (flux gating), excluding our former hypothesis that it essentially demands a down‐to‐up conformational change (revised Figure 6).